# Amyotrophic lateral sclerosis and parkinsonism in Papua, Indonesia: 2001–2012 survey results

Kiyohito Okumiya,[1,2] Taizo Wada,[2] Michiko Fujisawa,[2] Masayuki Ishine,[3] Eva Garcia del Saz,[4] Yutaka Hirata,[5] Shigeki Kuzuhara,[6] Yasumasa Kokubo,[7] Harumichi Seguchi,[8] Ryota Sakamoto,[9] Indrajaya Manuaba,[10] Paulina Watofa,[11] Andreas L Rantetampang,[12] Kozo Matsubayashi[2]

## ABSTRACT

**Objective:** Only one previous follow-up study of amyotrophic lateral sclerosis (ALS) and parkinsonism in Papua, Indonesia has been carried out since a survey undertaken in 1962–1981 by Gajdusek and colleagues. Therefore, to clarify the clinical epidemiology of ALS and parkinsonism in the southern coastal region of Papua, the clinical characteristics and prevalence of the diseases in this region were examined and assessed.

**Methods:** Cases of ALS and parkinsonism were clinically examined during a 2001–2012 survey in Bade and other villages along the Ia, Edera, Dumut and Obaa rivers in Papua, Indonesia. Possible, probable and definite ALS was diagnosed clinically by certified neurologists based on El Escorial criteria. The criteria for a diagnosis of parkinsonism were the presence of at least two of the four following signs: tremor, rigidity, bradykinesia and postural impairment with a progressive course.

**Results:** During the survey, 46 cases of ALS and/or parkinsonism were diagnosed within a population range of 7000 (2001–2002) to 13 900 (2007–2012). The 46 cases consisted of 17 probable-definite cases of ALS, including three with cognitive impairment (CI), 13 cases of overlapping possible, probable or definite ALS and parkinsonism, including five with CI, and 16 cases of parkinsonism, including one with CI. The crude point prevalence rate of pure ALS was estimated to be at least 73 (95% CI 0 to 156) to 133 (27 to 240)/100 000 people and that of overlapping ALS and parkinsonism at least 53 (0 to 126) to 98 (2 to 193)/100 000 in 2007, or 2010 in some regions.

**Conclusions:** While the prevalence of ALS in Papua has decreased over the past ~30–35 years, it remains higher than the global average. There was a high prevalence of overlapping ALS, parkinsonism and CI, which has also been previously reported in Guam and Kii.

For numbered affiliations see end of article.

**Correspondence to**
Dr Kiyohito Okumiya;
okumiyak@chikyu.ac.jp

## Strengths and limitations of this study

- This study is a unique epidemiological survey of neurodegenerative diseases from 2001 to 2012 in Papua, Indonesia, an area with one of the highest incidences of amyotrophic lateral sclerosis (ALS) in the world.
- This study recognised significant overlap of ALS with parkinsonism and cognitive impairment.
- This study was based only on clinical findings, so it is limited by lack of electromyogram data, DNA analysis, or any autopsy data from this population, without which it is difficult to determine how these patients fit into sporadic ALS or Parkinson's disease and how they compare with ALS/parkinsonism-dementia complex on Guam and in the Kii peninsula.

sclerosis (ALS), 18 cases of parkinsonism, and 18 cases of poliomyeloradiculitis (PMR) in the southern coastal area of Papua (former Irian Jaya, Indonesia), which had a population of 7000. The high incidence of ALS and parkinsonism in Papua was concentrated around the Ia River.[1 2] Along with Guam and Kii in Japan, Papua (Indonesia) was considered to have the highest incidence of ALS in the world.[1–7] However, the high incidence of ALS and parkinsonism-dementia complex (PDC) in Guam and Kii is reported to no longer exist.[8 9] Furthermore, in 1987, Spencer et al reported only two cases of ALS and three cases of parkinsonism in the villages along the Ia River. In 1990, no cases of ALS and three cases of parkinsonism were reported along the same river; thus, it was suspected that ALS has declined or even disappeared in Papua.[10] No follow-up surveys of ALS and parkinsonism have been conducted in Papua since 1991.

To clarify the clinical epidemiological characteristics of ALS and parkinsonism in

## INTRODUCTION

Between 1962 and 1980, Gajdusek et al reported 97 cases of amyotrophic lateral

the southern coastal region of Papua, we conducted a survey of neurodegenerative diseases documented between 2001 and 2012.

## METHODS

The survey sites for 2001 and 2002 were the villages of Kepi (population 800), Emete (800), Ogoto (400), Senggo (2000) and Bade (3000) for a total population of 7000.[11] [12] Between 2007 and 2012, the survey sites were the villages along four rivers, namely Bade on the Digul River (population 3800 people), Mur on the Mappi River (1500), the eight villages along the Ia River (4500), and the 12 villages along the Edera and Dumut rivers (4100 people) for a total population of 13 900. The eight villages along the Ia River were Gimikya (800), Geturki (500), Homlikya (700), Kobeta (600), Asset (600), Osso (300), Bosma (500) and Ogorito (500). The 12 villages along the Edera and Dumut rivers

were Benggo (200), Kokoya (500), Sahapikya (400), Harapan (400), Yibin (300), Mopio (300), Pies (300), Memes (300), Habeske (300), Muya (300), Getio (200) and Sien (600) (figure 1). At least two of six Japanese-certified neurologists (KO, MF, YH, SK, YK and KM) visited each village with a local neurologist (IM).

The village leaders and medical staff from the primary healthcare centre in each village agreed to take part in the survey, which was conducted with the permission of the Department of Health of the Papua Provincial Government in cooperation with Cenderawasih University. The purpose of the neurological examinations was explained to all villagers. All the subjects volunteered to participate after announcement of the availability of neurological check-ups, which were performed in the community health centres or in homes at each field site. While this was not a complete inventory survey, we asked each medical doctor or healthcare worker and the village leader to summon all patients

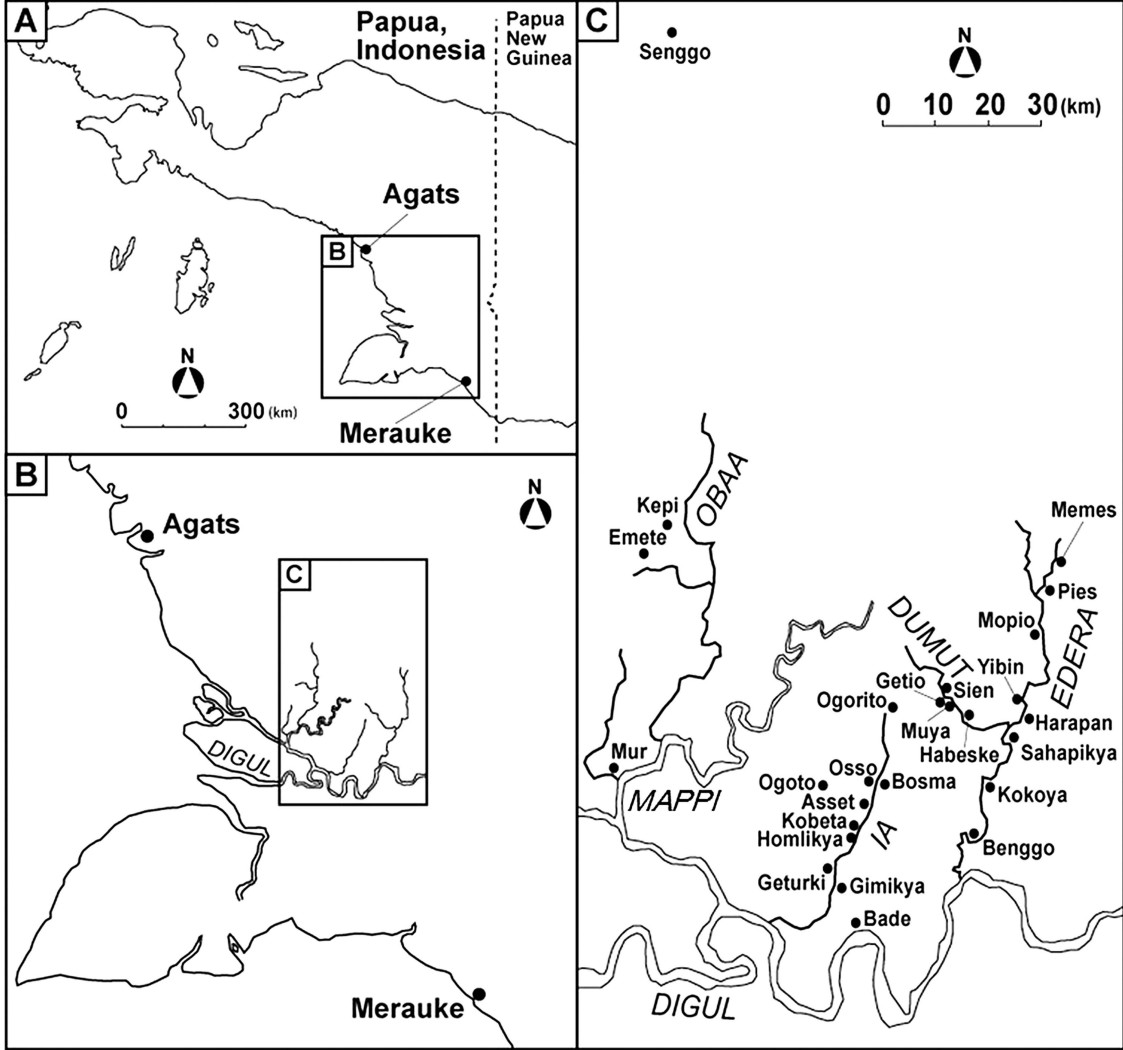

**Figure 1** Map of the field sites. (A) Large-scale map of the southern coastal plain in Papua, Indonesia. (B) Smaller scale map of the southern coastal plain in Papua, Indonesia showing rivers. (C) Detailed map showing villages in the southern coastal area in Papua in which all confirmed cases of amyotrophic lateral sclerosis and/or parkinsonism with/without cognitive impairment were seen from 2001 to 2012.

with neurological signs or symptoms, including muscle weakness, gait disturbance, tremor, bradykinesia or cognitive impairment (CI), in each village. All patients who were summoned also volunteered to participate in the study. Indonesian collaborators and those co-authors (an English teacher in Bade Senior High School, Indonesian neurologist IM, and staff from Cenderawasih University) who spoke both English and Indonesian, local people who spoke both Indonesian and the local language in each village, and co-author EG, who spoke English, Indonesian and Japanese, served as translators. Written informed consent was obtained from all study participants. For those participants who could not read/write, verbal communication was carried out with the support of their family members and the translators. All participants agreed to undergo neurological examinations, and none of the patients declined annual re-examination. The survey was approved by the ethics committees of the Research Institute for Humanity and Nature in Japan and Cenderawasih University in Papua, Indonesia.

Among the many neurological and non-neurological cases encountered, diagnoses of ALS, parkinsonism and CI were based on clinical examinations and discussions among the participating neurologists. ALS was classified as 'definite', 'probable', 'possible' or 'suspected' based on combinations of upper motor neuron (UMN) and lower motor neuron (LMN) signs according to the El Escorial criteria of the World Federation of Neurology.[13] Probable and definite ALS are defined as pure ALS in this report. A diagnosis of parkinsonism required the presence of at least two of the following four signs: tremor, rigidity, bradykinesia and postural impairment with a progressive course without evidence or history of vascular accidents or a history of taking drugs known to induce parkinsonism. The degree of disability of patients with parkinsonism was classified according to the scale of Hoehn and Yahr.[14] Overlapping of possible, probable or definite ALS with parkinsonism was defined as ALS-parkinsonism. CI was diagnosed when loss of memory or impairments of language usage, praxis or executive functions were identified during interview and clinical examination by neurologists. As cognitive functional tests such as the mini-mental state examination were not performed for all of the participants, patients with functional disability complications attributable to CI in activities of daily living or in their livelihood were screened during interview with the patients and their family members. PMR was evaluated using the definition of Gajdusek et al[1 2] that PMR is a subacute paralytic condition reminiscent of Landry–Guillain–Barré syndrome, which combines radicular, or perhaps neuritic, elements with more acute onset and sometimes asymmetric paralysis. Neurological cases with overlapping cerebellar signs (two cases) were excluded from this study.

In Bade and the villages along the Ia, Edera and Dumut rivers, new cases were identified and previously diagnosed cases were followed up and assessed neurologically and documented in 2007, 2008, 2010, 2011 and 2012. Years of death were confirmed, and the durations between the subjective onset of illness and the last survey or death were recorded as durations of subjective illness.

The size of the populations of Bade and the villages along the Ia, Edera and Dumut rivers were determined from a report by the public office of Edera district in 2006.[15] We were informed by the village leaders of the populations of other villages in the survey for the period 2001–2002.

## RESULTS

### Clinical types of ALS and/or parkinsonism

We identified 46 cases with signs of ALS and/or parkinsonism, including 17 cases with pure ALS (table 1), 13 with ALS-parkinsonism (table 2) and 16 with parkinsonism (table 3). We found no case of pure or dominant dementia and PMR consistent with the definition by Gajdusek et al; similarly, no cases of PMR were found by Spencer et al in 1987 or 1990.[10] Relevant information about the cases, including neurological signs, age at last survey, age and year of onset, disease duration, certified death year and ethnicity, is shown in tables 1–3. The male-to-female sex distribution was 10:7 (male 59%) for ALS, 11:2 (85%) for ALS-parkinsonism and 8:8 (50%) for parkinsonism. The mean age at last survey was 51.3 years for ALS, 52.9 years for ALS-parkinsonism and 51.6 years for parkinsonism. The mean age at onset was 46.2 years (range 11–68 years) for pure ALS, 49.5 years (11–70 years) for ALS-parkinsonism and 45.8 years (27–64 years) for parkinsonism. The mean duration of subjective illness was 7.0 years (2–20 years) for pure ALS, 6.3 years (2–17 years) for ALS-parkinsonism and 6.3 years (2–14 years) for parkinsonism. We observed overlapping CI in three (18%) of the 17 cases of pure ALS, five (38%) of the 13 cases of ALS-parkinsonism, and one (6%) of the 16 cases of parkinsonism.

During the follow-up surveys of the cases in Bade and along the Ia and Edera rivers, we observed the following changes in diagnosis: one case from possible ALS to probable ALS (case ALS 10); two cases from parkinsonism to ALS-parkinsonism (cases ALS-P 2 and ALS-P 3), and one case from probable ALS to definite ALS (case ALS 16) complicated with CI. We also identified an instance of a shared family history of parkinsonism (P 6) and ALS-parkinsonism (ALS-P 1). There were family histories of neurological disorders in seven of 17 cases (41%) of pure ALS, two (15%) of 13 cases of ALS-parkinsonism, and six (38%) of 16 cases of parkinsonism (tables 1–3).

### Cases of ALS and parkinsonism in Bade and the villages along the Ia, Edera and Dumut rivers between 2007 and 2012

Between 2007 and 2012, we identified four cases of pure ALS, two cases of ALS-parkinsonism, and one case of

**Table 1** Relevant characteristics and neurological signs in cases with pure ALS (n=17) with/without cognitive impairment in Papua, 2001–2012

| Case number | ALS criteria | Survey year | Village | Age | Sex | Onset (age/year) | Duration (years) | Death (year) | Tribe (origin) | UMN signs DTR U/L | BS | LMN signs MA U/L | Fasc U/L | Bulbar signs TA | Dys | Parkinsonian sign T U/L | R U/L | BK | PI | Cognitive impairment | Family history |
|---|---|---|---|---|---|---|---|---|---|---|---|---|---|---|---|---|---|---|---|---|---|
| ALS 1 | Definite | 2001 | Bade | 35 | M | NC | NC | NC | Auyu | 3/3 | + | ++/++ | +/+ | ++ | − | − | − | − | − | − | − |
| ALS 2 | Definite | 2001 | Ogoto | 43 | F | NC | NC | 2002 | Auyu | 3/3 | + | ++/++ | +/+ | ++ | ++ | − | − | − | − | − | − |
| ALS 3 | Definite | 2002 | Senggo | 50 | M | 48/'00 | 2 | NC | Citak | 3/3 | + | ++/++ | ±/± | ± | + | − | − | − | − | − | 1 |
| ALS 4 | Definite | 2007 | Homilikia | 53 | M | 51/'05 | 2 | 2007 | Auyu | 3/2 | + | ++/++ | +/+ | − | − | − | − | − | − | − | − |
| ALS 5 | Definite | 2007/'08 | Bade | 51 | M | 49/'06 | 3 | 2009 | Auyu | 3/4 | + | +/− | −/− | − | + | − | − | − | − | − | − |
| ALS 6 | Definite | 2008 | Bade | 21 | M | 11/'98 | 11 | 2009 | Auyu | 4/4 | + | ++/++ | −/− | − | − | − | − | − | − | − | 2 |
| ALS 7 | Definite | 2007/'08/'10 | Bade | 70 | F | 57/'97 | 13 | 2010 | Maluku | 3/3 | + | ++/++ | +/− | − | + | − | − | − | − | − | 3 |
| ALS 8 | Probable | 2011 | Geturki | 53 | M | 48/'06 | 5 | Alive | Auyu | 3/3 | − | +/+ | −/+ | − | − | − | − | − | − | − | − |
| ALS 9 | Definite | 2011 | Muya | 41 | F | 36/'06 | 6 | 2012 | Auyu | 3/3 | + | +/+ | −/− | − | − | − | − | − | − | − | − |
| ALS 10 | Probable | 2007/'10/'11/'12 | Osso | 65 | F | 59/'06 | 6 | Alive | Auyu | 3/4 | + | +/− | −/− | + | − | − | − | − | − | − | 4 |
| ALS 11 | Probable | 2007/'08/'10/'12 | Gimikya | 32 | M | 12/'92 | 20 | Alive | Auyu | 4/4 | + | ++/+ | ++/− | ± | − | − | +/− | − | − | − | 5 |
| ALS 12 | Definite | 2010/'11/'12 | Bosma | 64 | M | 53/'01 | 11 | Alive | Auyu | 3/4 | + | ++/++ | +/− | ++ | − | − | − | − | − | − | 6 |
| ALS 13 | Definite | 2012 | Asset | 70 | F | 68/'10 | 2 | Alive | Auyu | 3/3 | − | +/+ | −/− | − | − | − | − | − | − | − | − |
| ALS 14 | Probable | 2012 | Pies | 65 | F | 60/'07 | 5 | Alive | Auyu | 3/4 | − | +/+ | −/− | − | − | − | − | − | − | − | − |
| ALS 15 | Definite | 2007 | Bade | 64 | F | 62/'05 | 2 | 2007 | Maluku | 3/3 | + | ++/++ | −/− | − | + | − | − | − | − | + | 7 |
| ALS 16 | Definite | 2008/'10/'11 | Yibin | 25 | M | 13/'99 | 13 | 2012 | Auyu | 3/4 | + | ++/+ | ++/++ | − | − | − | − | − | − | + | − |
| ALS 17 | Definite | 2012 | Ogorito | 70 | M | 66/'08 | 4 | Alive | Auyu | 3/3 | + | +/+ | +/+ | − | − | − | − | − | − | + | − |

Age: at the time of final survey; Alive: alive in 2012; Deep tendon reflexes: −, absent; 1, hypoactive; 2, normal; 3, hyperactive; 4, markedly hyperactive with ankle clonus; Duration: between subjective onset of illness and last survey or death; Onset: subjective onset of illness.

Other signs are classified as: −, absent; ±, questionably present; +, present; ++, markedly present.

Family history: 1, cousin (possible ALS); 2, grandfather (NC); 3, sister (ALS 15); 4, cousin's daughter (P 16); 5, cousin (NC); 6, father (NC); 7, sister (ALS 7).

ALS, amyotrophic lateral sclerosis; BK, bradykinesia; BS, Babinski sign; DTR, deep tendon reflex; Dys, dysphagia/dysarthria; F, female; Fasc, fasciculation; LMN, lower motor neuron signs; M, male; MA, muscle atrophy with weakness; NC, not confirmed by examination; P, parkinsonism; PI, postural impairment; R, rigidity; T, tremor; TA, tongue atrophy; U/L: upper limbs/lower limbs; UMN, upper motor neuron signs; Yahr stage, the scale of Hoehn and Yahr.

**Table 2** Relevant characteristics and neurological signs in cases with overlapping ALS and parkinsonism (n=13) with/without cognitive impairment in Papua, 2001–2012

| Case number | ALS criteria, Yahr stage | Survey year | Village | Age | Sex | Onset (age/year) | Duration (year) | Death (year) | Tribe (origin) | UMN signs DTR U/L | BS | LMN signs MA U/L | Fasc U/L | Bulbar signs TA | Dys | Parkinsonian sign T U/L | R U/L | BK | PI | Cognitive impairment | Family history |
|---|---|---|---|---|---|---|---|---|---|---|---|---|---|---|---|---|---|---|---|---|---|
| ALS-P 1 | Possible, IV | 2008 | Pies | 65 | F | 60/'03 | 6 | 2009 | Auyu | 3/3 | + | +/− | −/− | − | − | +/+ | ++/++ | ++ | + | − | 8 |
| ALS-P 2 | Possible, III | 2010/'11 | Ogorito | 76 | M | 70/'05 | 6 | 2011 | Auyu | 3/3 | + | +/± | −/− | − | − | +/+ | +/+ | + | + | − | − |
| ALS-P 3 | Possible, III | 2010/'11/'12 | Homilikia | 69 | M | 65/'08 | 4 | Alive | Auyu | 3/2 | + | +/− | −/− | − | − | +/+ | +/+ | + | + | − | − |
| ALS-P 4 | Definite, III | 2008 | Bade | 34 | M | 18/'92 | 17 | 2009 | Muyu | 3/3 | + | +/− | −/− | ++ | ++ | − | +/+ | + | + | + | 9 |
| ALS-P 5 | Definite, III | 2012 | Sien | 60 | M | 58/'10 | 2 | Alive | Auyu | 3/3 | + | +/+ | −/− | − | − | − | +/+ | + | + | + | − |
| ALS-P 6 | Definite, III | 2012 | Benggo | 50 | M | 48/'10 | 2 | Alive | Auyu | 2/2 | + | +/+ | −/− | + | + | − | +/− | + | + | + | − |
| ALS-P 7 | Possible, II | 2001 | Bade | 36 | M | NC | NC | NC | Auyu | 3/3 | + | −/− | −/− | − | − | + | + | + | − | − | − |
| ALS-P 8 | Possible, II | 2008/'10 | Bade | 23 | M | 11/'98 | 12 | Alive | Auyu | 3/4 | + | −/− | −/− | − | − | +/+ | +/± | + | − | − | − |
| ALS-P 9 | Possible, III | 2010/'11 | Ogorito | 76 | M | 70/'05 | 6 | 2011 | Auyu | 3/3 | + | −/− | −/− | − | − | +/− | +/+ | + | + | − | − |
| ALS-P 10 | Possible, III | 2008/'10/'11 | Yibin | 52 | M | 44/'03 | 8 | Alive | Auyu | 3/3 | + | −/− | −/− | − | − | +/− | +/+ | + | + | − | − |
| ALS-P 11 | Possible, III | 2011 | Getio | 55 | F | 53/'09 | 2 | 2011 | Auyu | 3/3 | + | ±/− | −/− | − | − | +/+ | +/+ | + | + | − | − |
| ALS-P 12 | Possible, IV | 2001 | Bade | 40 | M | NC | NC | NC | Auyu | 3/3 | + | −/− | −/− | − | + | + | + | + | + | + | − |
| ALS-P 13 | Possible, III | 2012 | Bosma | 52 | M | 48/'08 | 4 | Alive | Auyu | 3/4 | + | −/− | −/− | − | − | − | ++/+ | + | + | + | − |

Age: at the time of final survey; Alive: alive in 2012; Deep tendon reflexes: −, absent; 1, hypoactive; 2, normal; 3, hyperactive; 4, markedly hyperactive with ankle clonus; Duration: between subjective onset of illness and last survey or death; Onset: subjective onset of illness.
Other signs are classified as: −, absent; ±, questionably present; +, present; ++, markedly present.
Family history: 8, sister (P 6) and cousin (NC); 9, father (NC).
ALS, amyotrophic lateral sclerosis; BK, bradykinesia; BS, Babinski sign; DTR, deep tendon reflex; Dys, dysphagia/dysarthria; F, female; Fasc, fasciculation; LMN, lower motor neuron signs; M, male; MA, muscle atrophy with weakness; NC, not confirmed by examination; P, parkinsonism; PI, postural impairment; R, rigidity; T, tremor; TA, tongue atrophy; U/L, upper limbs/lower limbs; UMN, upper motor neuron signs; Yahr stage, the scale of Hoehn and Yahr.

**Table 3** Relevant characteristics and neurological signs in cases with parkinsonism with/without cognitive impairment (n=16) in Papua, 2001–2012

| Case number | Yahr stage | Survey year | Village | Age | Sex | Onset (age/year) | Duration (years) | Death (year) | Ethnicity | UMN signs DTR U/L | BS | LMN signs MA U/L | Fasc U/L | Bulbar signs TA | Dys | Parkinsonian sign T U/L | R U/L | BK | PI | Cognitive impairment | Family history |
|---|---|---|---|---|---|---|---|---|---|---|---|---|---|---|---|---|---|---|---|---|---|
| P 1 | Yahr IV | 2001 | Kepi | 50 | M | NC | NC | NC | Jakai | 2/2 | – | –/– | –/– | – | – | +/+ | +/+ | ++ | + | – | – |
| P 2 | Yahr II | 2001 | Kepi | 50 | M | NC | NC | NC | Jakai | 2/2 | – | –/– | –/– | – | – | + | + | + | – | – | – |
| P 3 | Yahr IV | 2002 | Emete | 52 | M | 49/'99 | 3 | NC | Jakai | 2/2 | – | –/– | –/– | – | – | + | + | + | + | – | – |
| P 4 | Yahr II | 2002 | Senggo | 41 | M | 36/'97 | 5 | NC | Citak | 2/2 | – | –/– | –/– | – | – | + | + | – | – | – | – |
| P 5 | Yahr I | 2007 | Mur | 65 | F | 61/'03 | 4 | NC | Jakai | 2/3 | – | –/– | –/– | – | – | +/– | +/– | – | – | – | 10 |
| P 6 | Yahr IV | 2008 | Pies | 60 | F | 55/'03 | 6 | 2009 | Auyu | 2/2 | – | –/– | –/– | – | – | +/+ | +/+ | ++ | + | – | 11 |
| P 7 | Yahr IV | 2008/'10 | Mopio | 52 | F | 47/'05 | 5 | Alive | Auyu | 3/3 | – | –/– | –/– | – | – | ++/+ | +/+ | ++ | + | – | – |
| P 8 | Yahr II | 2011 | Sien | 57 | F | 49/'03 | 9 | 2012 | Auyu | 3/3 | – | –/– | –/– | – | – | +/+ | +/+ | + | – | – | 12 |
| P 9 | Yahr IV | 2007/'08/'10/'12 | Bade | 46 | F | 40/'06 | 6 | Alive | Auyu | 3/3 | – | –/– | –/– | – | – | +/+ | ++/+ | + | + | – | 13 |
| P 10 | Yahr II | 2011/'12 | Asset | 38 | M | 27/'01 | 11 | Alive | Auyu | 3/3 | – | –/– | –/– | – | – | +/– | +/– | + | – | – | – |
| P 11 | Yahr III | 2011/'12 | Sien | 54 | F | 40/'98 | 14 | Alive | Auyu | 3/3 | – | –/– | –/– | – | – | ++/+ | ++/++ | + | + | – | – |
| P 12 | Yahr II | 2011/'12 | Memes | 66 | F | 64/'10 | 2 | Alive | Auyu | 3/3 | – | –/– | –/– | – | – | +/+ | +/+ | + | – | – | 14 |
| P 13 | Yahr II | 2012 | Harapan | 60 | M | 57/'09 | 3 | Alive | Auyu | 2/2 | – | –/– | –/– | – | – | – | +/+ | – | – | – | – |
| P 14 | Yahr II | 2007 | Mur | 50 | M | 38/'95 | 12 | NC | Auyu | 2/4 | + | –/– | –/– | – | – | + | + | + | – | – | – |
| P 15 | Yahr I | 2007 | Homilikia | 50 | M | NC | NC | Alive | Auyu | 2/3 | + | –/– | –/– | – | – | +/– | +/± | – | – | – | – |
| P 16 | Yahr III | 2011 | Osso | 35 | F | 33/'09 | 2 | 2011 | Auyu | 3/2 | – | –/– | –/– | – | – | +/+ | +/+ | + | + | + | 15 |

Age: at the time of final survey; Alive: alive in 2012; Deep tendon reflexes: –, absent; 1, hypoactive; 2, normal; 3, hyperactive; 4, markedly hyperactive with ankle clonus; Duration: between subjective onset of illness and last survey or death; Onset: subjective onset of illness.
Other signs are classified as: –, absent; ±, questionably present; +, present; ++, markedly present.
Family history: 10, son (NC); 11, sister (ALS-P 1) and cousin (NC); 12, mother and father (NC); 13, grandmother (NC); 14, mother (NC); 15, mother's cousin (ALS 10).
ALS, amyotrophic lateral sclerosis; BK, bradykinesia; BS, Babinski sign; DTR, deep tendon reflex; Dys, dysphagia/dysarthria; F, female; Fasc, fasciculation; LMN, lower motor neuron signs; M, male; MA, muscle atrophy with weakness; NC, not confirmed by examination; P, parkinsonism; PI, postural impairment; R, rigidity; T, tremor; TA, tongue atrophy; U/L, upper limbs/lower limbs; UMN, upper motor neuron signs; Yahr stage, the scale of Hoehn and Yahr.

parkinsonism in Bade (population 3800). In addition, we found seven cases of pure ALS, four cases of ALS-parkinsonism, and three cases of parkinsonism in the villages (population 4500 people) along the Ia river. Between 2008 and 2012, we identified three cases of pure ALS, five cases of ALS-parkinsonism, and six cases of parkinsonism in the villages (population 4100 people) along the Edera and Dumut rivers (table 4).

In the 2007 survey, there were four active cases of pure ALS (crude point prevalence rate 105 (95% CI 2 to 208)/100 000 people), two cases of ALS-parkinsonism (53 (0 to 126)/100 000 people) and one case of parkinsonism (26 (0 to 78)/100 000 people) in Bade. In 2010, there were six active cases of pure ALS (133 (27 to 240)/100 000 people), four cases of ALS-parkinsonism (89 (2 to 176)/100 000 people) and three cases of

parkinsonism (67 (0 to 142)/100 000 people) in the villages along the Ia River. In 2010, there were three active cases of pure ALS (73 (0 to 156)/100 000 people), four cases of ALS-parkinsonism (98 (2 to 193)/100 000 people) and five cases of parkinsonism (122 (15 to 229)/100 000 people) in the villages along the Edera and Dumut rivers (table 4).

In the survey during the period 2007–2012, the number of each clinical type (ALS, ALS-parkinsonism and parkinsonism, respectively) was determined for Bade (4, 4 and 1), the villages along the Ia River, namely Ogorito (1, 2 and 0), Bosma (1, 1 and 0), Homlikya (1, 1 and 1), Gimikya (1, 1 and 0), Geturki (1, 0 and 0), Asset (1, 0 and 1) and Osso (1, 0 and 1), and the villages along the Edera and Dumut rivers, namely Yibin (1, 1 and 0), Sien (0, 1 and 2), Pies (1, 0 and 1),

**Table 4** Cases of amyotrophic lateral sclerosis (ALS) and parkinsonism in the field sites

| | ALS criteria | Case no. | Crude point prevalence rate (95% CI) among ALS/ ALS- parkinsonism/parkinsonism (2007* or 2010†) |
|---|---|---|---|
| Bade 2007–2012 (population 3800) | | | |
| ALS | Definite | ALS 5* | 105 (2 to 208)/100 000 in 2007* |
| (n=4) | Definite | ALS 6* | |
| | Definite | ALS 7* | |
| | Definite | ALS 15* | |
| ALS-parkinsonism | Definite | ALS-P 4* | 53 (0 to 126)/100 000 in 2007* |
| (n=2) | Possible | ALS-P 8* | |
| Parkinsonism (n=1) | | P 9* | 26 (0 to 78)/100 000 in 2007* |
| Ia River 2007–2012 (population 4500) | | | |
| ALS | Definite | ALS 4 | 133(27 to 240)/100 000 in 2010† |
| (n=7) | Probable | ALS 8† | |
| | Probable | ALS 10† | |
| | Probable | ALS 11† | |
| | Definite | ALS 12† | |
| | Definite | ALS 13† | |
| | Definite | ALS 17† | |
| ALS-parkinsonism | Possible | ALS-P 2† | 89 (2 to 176)/100 000 in 2010† |
| (n=4) | Possible | ALS-P 3† | |
| | Possible | ALS-P 9† | |
| | Possible | ALS-P 13† | |
| Parkinsonism | | P 10† | 67 (0 to 142)/100 000 in 2010† |
| (n=3) | | P 15† | |
| | | P 16† | |
| Edera and Dumut rivers 2008–2012 (population 4100) | | | |
| ALS | Definite | ALS 9† | 73 (0–156)/100 000 in 2010† |
| (n=3) | Probable | ALS 14† | |
| | Definite | ALS 16† | |
| ALS-parkinsonism | Possible | ALS-P 1 | 98 (2 to 193)/100 000 in 2010† |
| (n=5) | Definite | ALS-P 5† | |
| | Definite | ALS-P 6† | |
| | Possible | ALS-P 10† | |
| | Possible | ALS-P 11† | |
| Parkinsonism | | P 6 | 122 (15 to 229)/100 000 in 2010† |
| (n=6) | | P 7† | |
| | | P 8† | |
| | | P 11† | |
| | | P 12† | |
| | | P 13† | |

*Active cases in 2007 in Bade.
†Active cases in 2010 in the villages along the Ia, Edera and Dumut rivers.

Benggo (0, 1 and 0), Harapan (0, 0 and 1), Mopio (0, 0 and 1), Memes (0, 0 and 1), Muya (1, 0 and 0) and Getio (0, 1 and 0).

## Case reports

### Case ALS 12

A 64-year-old (in 2012) man of the Auyu tribe, a resident of Bosma along the Ia River, in 2010 had a 9-year history of progressive gait disturbance and motor weakness. These symptoms were found to have progressed based on the findings from follow-up check-ups between 2011 and 2012. Neurological examination revealed UMN and LMN signs, including a positive Babinski sign, hyperreflexia with ankle clonus, muscle atrophy, weakness in the tongue and upper and lower limbs bilaterally, and fasciculation in the upper limbs and tongue. He had no CI or sensory disturbance. We diagnosed him as having definite ALS (figure 2A–D). The subject had ALS, and his brother reported that their father had showed tremor and gait disturbance, which were interpreted to be consistent with parkinsonism.

### Case ALS-CI 17

A 70-year-old (in 2012) man of the Auyu tribe, a resident of Ogorito along the Ia River, had a history of progressive memory loss for 4 years and slow gait for 2 years.

Neurological examination revealed UMN and LMN signs, including a positive Babinski sign, hyperreflexia in the jaw jerk and the bilateral upper and lower limbs, and muscle weakness with atrophy and fasciculation in the distal upper and lower limbs bilaterally. He had fasciculation in the tongue and respiratory muscle weakness with a low oxygen saturation level ($SpO_2$ 86%). He also had memory disturbances accompanied by positive snout and palmomental reflexes and difficulty with mental activities of daily living. He had no sensory disturbance. We diagnosed him as having definite ALS with CI (figure 3A).

### Case ALS-P 1

A 65-year-old (in 2008) woman of the Auyu tribe, a resident of Pies along the Edera River, had a 5-year history of progressive motor weakness and gait disturbance. She could not walk without support. Neurological examination revealed a resting tremor, cogwheel rigidity in the upper and lower limbs, bradykinesia throughout, postural impairment, UMN and LMN signs with a positive Babinski sign, hyperreflexia in the upper and lower limbs bilaterally, and thenar and hypothenar muscle atrophies. She had no CI or sensory disturbance. We diagnosed her as having ALS-parkinsonism and, more specifically, severe parkinsonism (Hoehn and Yahr IV)

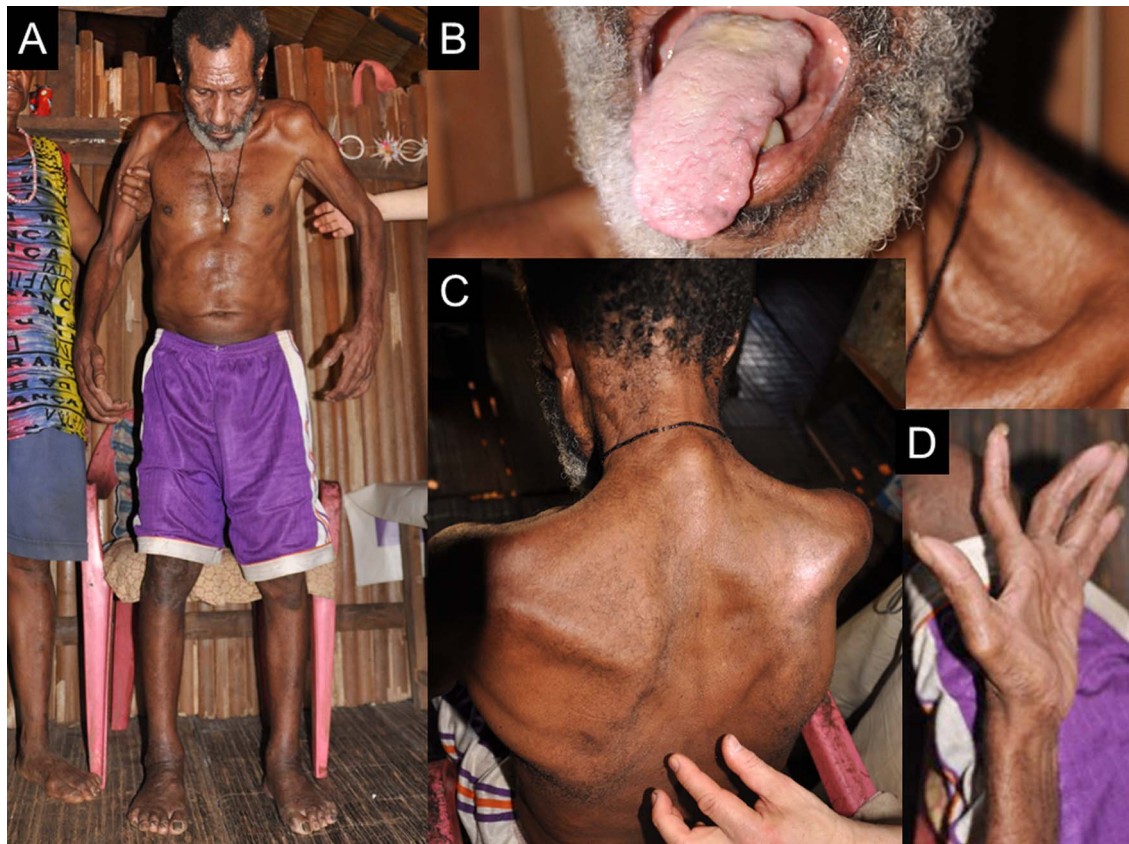

**Figure 2**  Patient with amyotrophic lateral sclerosis (ALS) (case ALS 12). (A) The patient needs assistance from family members to stand. (B) Advanced atrophy of the tongue. (C) There is upper limb girdle and truncal muscle atrophy with a positive Babinski sign. (D) Advanced thenar muscle atrophy.

simultaneously complicated by motor neuron signs indicative of possible ALS. She died in 2009. Her younger sister was case P 6. The subject had ALS-parkinsonism, and her sister reported that their cousin showed tremor and gait disturbance, which were interpreted to be consistent with parkinsonism (figure 3B).

### Case ALS-P 2

A 76-year-old (in 2011) man of the Auyu tribe, a resident of Ogorito, had a 5-year history of tremor in both hands in 2010. Neurological examination revealed cogwheel rigidity in his upper and lower limbs, bradykinesia, and UMN signs with hyperreflexia in bilateral upper and lower limbs. We diagnosed him as having parkinsonism (Hoehn and Yahr II) in 2010. He was examined again in 2011, by which time his parkinsonism was complicated by postural impairment, muscle weakness and atrophy, and spasticity of the upper and lower limbs with Babinski signs. He had no CI or sensory disturbance. We diagnosed him as having parkinsonism (Hoehn and Yahr III) with possible ALS; parkinsonism was accompanied by ALS later. He died in 2011 of unknown causes.

We also diagnosed another case of parkinsonism (Hoehn and Yahr III) with possible ALS (ALS-P 3); parkinsonism was accompanied by ALS later.

### Case ALS-P-CI 6

A 50-year-old (in 2012) man of the Auyu tribe, a resident of Benggo along the Edera River, had a 2-year history of dysarthria. Neurological examination revealed UMN and LMN signs with a positive Babinski sign, muscle atrophy, and weakness in the tongue, proximal upper limb, shoulder and tibialis anterior muscles. He had cogwheel rigidity in the wrist and elbow, bradykinesia throughout, a positive Myerson sign, and impairment of coordination and postural reflexes. He had memory loss, bradyphrenia, slow speech, and positive snout and palmomental reflexes with difficulty performing activities of daily living. We diagnosed him as having definite ALS and parkinsonism (Hoehn and Yahr III) with CI (figure 3C).

### Case P 7

A 52-year-old (in 2010) woman of the Auyu tribe who lived in Mopio along the Edera River had a history of progressive gait disturbance with parkinsonian signs, including resting tremor, cogwheel rigidity, anterior bending, and bradykinesia for 3 years in 2008. In 2008, we diagnosed her as having parkinsonism (Hoehn and Yahr II). By the time of the follow-up survey in 2010, she could not stand by herself and had postural impairment with parkinsonian hands bilaterally. She had no UMN or LMN signs, no CI and no sensory disturbance. We diagnosed her as having parkinsonism (Hoehn and Yahr IV; figure 3D).

## DISCUSSION

Internal migration and migration from outside the province are the most significant causes of demographic, social and cultural change in Papua. The populations in the survey sites were composed of former hunter-gatherers who had lived in the forest but had been brought out of the forest and assembled in villages by Dutch colonialists. The villages along the Ia River were founded between 1937 and 1951.[10] An emigration programme called transmigration ('transmigrasi') started in Irian Jaya in 1964 and led to an increase in the number of local migrants and transmigrants from outside Irian Jaya, resulting in various changes, including in the use of food and medicine.[16][17] As Gadjusek et al found a high prevalence of ALS in the first survey in 1962, the neurodegenerative disease must have existed before the introduction of manufactured items.[1]

In the present survey, we performed full neurological examinations in 46 subjects with ALS and/or parkinsonism between 2001 and 2012. Among the subjects, 17 were probable-definite cases of pure ALS, 13 were overlapping ALS (possible, probable or definite) and parkinsonism, and 16 were parkinsonism. Nine cases of ALS and/or parkinsonism had the complication of CI. During the study period, the diagnosis in some patients changed from pure parkinsonism to ALS-parkinsonism, and their disease progressed in severity. Some subjects had both ALS and parkinsonism on their first examination. It is thus evident that overlapping ALS and parkinsonism with or without CI still exists in this region of Papua.

Gajdusek et al[1] reported diagnosing 97 cases (57 cases clinically confirmed and 40 cases not clinically confirmed) of ALS during the six surveys they conducted from 1974 to 1981. However, they reported the neurological findings of only 32 of these cases, in only 16 of whom the reported findings of UMN, LMN and parkinsonian signs justify a diagnosis of probable or definite ALS. The 16 cases comprise 12 cases of pure ALS and four cases of overlapping ALS-parkinsonism. Among the 32 ALS patients, three had a positive family history (9%): two had sisters with ALS, and one with ALS-parkinsonism had a family history of ALS. None of the ALS patients had a family history of parkinsonism. The authors also reported their neurological findings in 13 subjects in whom parkinsonism was diagnosed. They reported their findings from full neurological examinations for 10 of these parkinsonism cases. According to their reported neurological signs, these comprised eight patients with parkinsonism, including five with dementia, and two with overlapping ALS-parkinsonism with dementia.[1] Among the 13 patients with parkinsonism, three had a positive family history (23%). In summary, we believe that in the 26 cases reported by Gajdusek et al,[1] the full neurological examination findings comprised 12 pure ALS cases (probable or definite), six overlapping ALS-parkinsonism, and eight parkinsonism with/without dementia. Spencer et al reported two cases

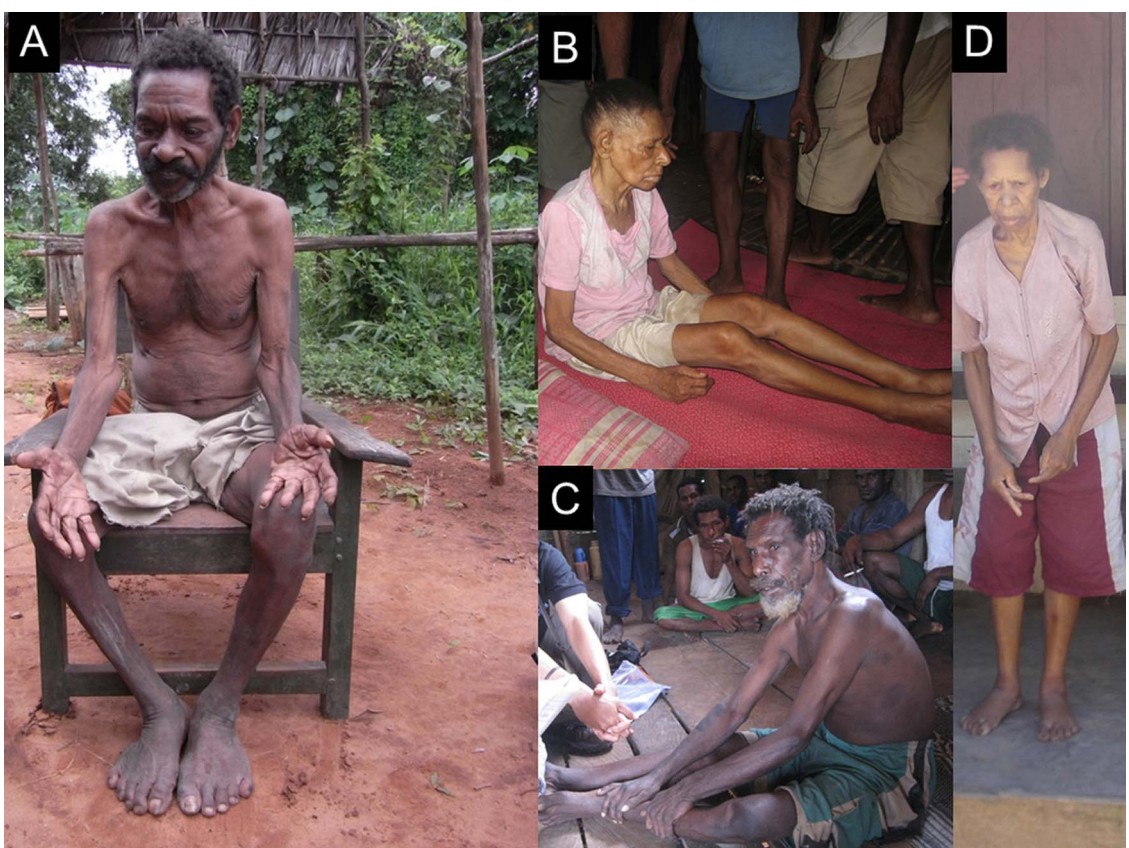

**Figure 3** Patients with amyotrophic lateral sclerosis (ALS), ALS-parkinsonism and parkinsonism. (A) Patient with ALS (case ALS-17) showing muscle atrophy in the distal upper and lower limbs. (B) Patient with ALS-parkinsonism (case ALS-P 1) showing muscle atrophy in distal upper limbs. (C) Patient with ALS-parkinsonism (case ALS-P 7) showing muscle atrophy in the proximal upper limb, shoulder and tibialis anterior. (D) Patient with parkinsonism (case P 7) showing typical parkinsonian hands and anterior bending posture. Parkinsonian hand: striatal deformities of the hand with abnormal postures that are common in patients with advanced Parkinson's disease.

of ALS and three cases of parkinsonism, including one with dementia, in their survey in 1987.[10] They also reported three subjects with parkinsonism overlapping UMN signs, including one with dementia in 1990.[10] We have here reported 17 cases of pure ALS (probable or definite), 13 cases of overlapping ALS-parkinsonism, and 16 cases of parkinsonism with or without CI in the 46 cases we examined fully neurologically. Thus, we identified many cases of overlapping ALS and parkinsonism (n=13, 28%) similarly to previous reports (n=6, 23% and n=3, 38%).[1 10] We also identified many cases with CI (n=9, 20%), also similarly to previous reports (n=7, 27% and n=2, 25%).[1 10] Functional disabilities attributable to CI were recognised in activities of daily living or in their livelihood in all cases with CI in this report.

Gajdusek *et al*[1] reported 13 cases of ALS and three cases of parkinsonism in 1975 in the villages along the Ia River (population 2000). The 13 ALS cases consisted of 10 clinically confirmed cases and three others which were not clinically confirmed. We compared the prevalence of probable-definite ALS and parkinsonism reported by Gajdusek *et al* with our more recent findings. The crude point prevalence of ALS consistent with

clinically probable to definite ALS (El Escorial criteria) was between 500 (95% CI 191 to 809)/100 000 people and 650 (298 to 1002)/100 000 people, while that of parkinsonism was 150 (0 to 320)/100 000 people in 1975 along the Ia River. In our 2010 survey (population 4500), we identified six probable-definite cases of ALS in this region (patients ALS 8, 10–13 and 17; 133 (27 to 240)/100 000 people) and seven of parkinsonism (possible ALS-P 2, 3, 9 and 13, and P 10, 15 and 16; 156 (40 to 271)/100 000 people).

Gajdusek *et al* reported eight cases of ALS and six cases of parkinsonism in 1975 in the villages along the Edera and Dumut rivers (population 1950). The eight cases of ALS consisted of seven clinically confirmed cases and one unconfirmed case. The crude point prevalence of ALS consistent with clinically probable to definite ALS (El Escorial criteria) was between 359 (94 to 624)/100 000 people and 410 (127 to 694)/100 000 people, while that of parkinsonism was 308 (62 to 554)/100 000 people in 1975 along the Edera and Dumut rivers. In our survey of this region in 2010 (population 4100), we identified five probable-definite cases of ALS (ALS 9, 14 and 16, and ALS-P 5 and 6; 122 (15 to 229)/100 000 people) and seven cases of parkinsonism

(possible ALS-P 10 and 11, and P 7, 8 and 11–13; 171 (44 to 297)/100 000 people; table 4). As this was not a complete inventory survey, the actual prevalence may be a little higher than our estimation.

The population of Bade increased nearly fourfold over 31 years (from 1000 in 1975 to 3800 in 2006), and those of villages along the Ia River (from 2000 to 4500) and the Edera and Dumut rivers (from 1950 to 4100) have increased twofold.[1 15] Only two of the 46 subjects with ALS and/or parkinsonism with/without CI in this survey were born outside Papua (in Maluku). The prevalence of the diseases among the local Papuan residents in this survey may be underestimated compared with the former report after taking into account the increase in the numbers of local migrants from inside Papua and transmigrants from outside Papua.[16–18]

As the median age and life expectancy at the field sites in Papua are estimated to be lower than the global median and fewer people may live beyond the age of 40 or 50 years,[18 19] the prevalence of the diseases in this survey may be underestimated compared with the global mean. As the median age and life expectancy in 1975 may be lower than in 2010,[20] the prevalence of the diseases in the report by Gajdusek et al might be underestimated compared with that in our survey.

The prevalence of ALS in this report was found to be three times lower than that found by Gajdusek et al. Even after taking into account the changes in population and age structure at the field sites, our findings show that the prevalence of ALS appears to have decreased over the past ~30–35 years since the report by Gajdusek et al.[1] This is consistent with the previous report by Spencer et al.[10] However, the prevalence is still much higher than the global mean (incidence ~2/100 000; prevalence ~6/100 000)[21–26]; in particular, the prevalence of overlapping ALS and parkinsonism is especially high in Papua.

In the report by Gajdusek et al,[1 2] who provided village-by-village data showing the number of cases of ALS and parkinsonism, there were many cases in the remote villages such as Bosma, Pies, Homlikya, Yibin, Sien and Ogorite, which were less visited and least developed. In this survey, many cases (three ALS/ALS-parkinsonism) were found in the remotest village of Ogorito along the Ia River. Two cases of ALS/ALS-parkinsonism were also found in Bosma, Homlikya and Yibin, one ALS and two cases of parkinsonism in Sien, and one case ALS and one of parkinsonism in Pies, which were all remote villages. While there were eight cases of ALS/ALS-parkinsonism in Bade and two in nearby Gimikya, none was reported by Gajdusek et al in those sites. This increase may have been brought about by local migrants from affected villages.

Overlapping ALS-parkinsonism and dementia has also been reported in patients with ALS/PDC in Guam and Kii, Japan.[4 27–33] Hirano et al provided clinicopathological evidence of 47 cases of PDC in Guam in 1961, comprising the following three groups: 30 cases of PDC;

eight cases of parkinsonism with dementia with clinical evidence of UMN impairment; and nine cases of parkinsonism with dementia with clinical evidence of UMN and LMN impairment. At least 15 of these cases (31.9%) had a family history of ALS, parkinsonism or both.[4] Kuzuhara and Kokubo et al surveyed neurological diseases in Kii, Japan in the 1990s and found a continuously high incidence of ALS and neuropathologically verified cases of PDC. ALS and PDC frequently affected the same individuals simultaneously and members of the same family. These authors reported 37 cases of ALS/PDC between 1996 and 2006, a high proportion of whom (78%) had a positive family history, as well as clinical examples of pure ALS, PDC and overlapping ALS and PDC. They also reported neuropathological changes common to cases of ALS and PDC in 12 cases of ALS/PDC, resembling those found in patients with ALS/PDC in Guam.[28–33] In our more recent study in Papua, as the previous studies in Kii, Japan and Guam, we identified many cases of pure ALS and ALS overlapping with parkinsonism with/without CI. The prevalence of CI was lower in our study sites than in Guam and Kii.[4 27–33] CI might have been underestimated, and mild CI or mild dementia might have been overlooked by our neurological examinations in the field.

Our study was based only on clinical findings, and limitations include lack of electromyogram data, DNA analysis data, or any autopsy data from this population. Without such data, it is difficult to determine how these patients fit into sporadic ALS or Parkinson disease and how they compare with ALS/PDC on Guam and in the Kii peninsula. However, we did identify many patients with overlapping ALS-parkinsonism in this study. The overlapping of ALS and/or parkinsonism with CI suggests that it may be the same disease entity as the ALS/PDC diagnosed in Guam and Kii.

In Bade, the villages along the Ia River, and the villages along the Edera and Dumut rivers, 53%, 51% and 50%, respectively, of the population were male. There was a high proportion of males especially among subjects with ALS-parkinsonism (85%) and ALS (59%) compared with pure parkinsonism (50%) in our report. The proportion of males was higher among the subjects with ALS (63%) and parkinsonism (76%) in the report by Gadjusek et al, in which three of the four ALS-parkinsonism cases were male (75%). The high risks for ALS and PDC among male subjects were recognised in Guam and Kii.[1 7 27]

Indigenous people of Papua still retain aspects of their traditional lifestyles, such as hunting, gathering, fishing and eating sago (from the trunk of *Metroxylon* sp.). They also drink water from shallow wells. However, with the spread of a market economy, recent increases in the numbers of rubber plantations and progress in transportation have been causing lifestyle changes in the villages in the southern coastal area of Papua. Since the diseases are declining as in Guam and Kii, the disease aetiology in Papua may be dominated by environmental factors.

Genetic, culture-related and/or family-specific factors may also play a role as evidenced by the family history of disease in 40% of the subjects in this survey in Papua and in 80% in Kii.

Various possible causal genetic and environmental factors for these conditions have been reported. The environmental factors of low concentrations of calcium and magnesium in drinking water were hypothesised.[1 34] There is a recent report showing an association between an increase in ALS incidence and a change in sources of drinking water in the Kii Peninsula.[35] Spencer et al[10] pointed out that this explanation was inconsistent with a declining prevalence of ALS in a sessile population dependent on an unchanging supply of river water, which this report may support.

Spencer et al[36] reported that cycad seed was used as a poultice in Papua as in Guam. In addition, epidemiological evidence showed that exposure to cycad-derived products was a risk factor for dementia, mild CI and PDC in Guam.[37] Although cycad had been suspected to be the causal factor of ALS in Kii,[38] people in Japan, including the Kii Peninsula, never ate cycad as a daily food, although a small amount of cycad flour in herbal medicine may have been consumed by a few people.[39] We did not examine the association between prior skin wounds and exposure to cycad poultice with the diseases. Skin change and pigmentary retinopathy were not examined in this survey.[40 41] Further studies are needed.

Genetic abnormalities previously established to be associated with familial or sporadic ALS, dementia, familial Parkinson disease and parkinsonism, were sought in the cases in Kii, but none were identified.[28 42] Hermosura et al[43] discovered a nuclear mutation in the calcium/magnesium membrane ion channel transient receptor potential melastatin 7 (*TRPM7*) gene in a subset of patients with ALS/PDC in Guam. They reported that a TRPM7 variant that is associated with altered sensitivity to magnesium may have contributed to the pathogenesis of ALS and PDC in two Guamanians. *TRPM7* was not found to be associated with ALS/PDC in the Kii peninsula of Japan.[44] Ishiura et al[45] recently discovered a *C9ORF72* repeat expansion in some cases of ALS in the Kii peninsula of Japan; this repeat expansion partly accounts for the high prevalence of ALS in this region. However, the responsible gene(s) remain unproven in almost all cases of ALS/PDC in both Guam and Kii.

Various environmental factors and multiple genetic factors are suggested to be underlying causes of these diseases, but these links are so far unproven.[33 46]

In Guam, although the incidence of pure ALS has declined markedly, the prevalence of PDC is still high.[47] The prevalence of dementia among elderly Chamorros (the earliest known inhabitants of the Mariana Islands, which include Guam) is also reported to be relatively high.[48] In addition, in Kii the incidence of pure ALS has declined markedly, although the prevalence of PDC and ALS/PDC or ALS with dementia has recently increased.[30 33] These changing patterns of ALS/PDC incidence in Guam and Kii might be caused by changes in environmental and socioeconomic factors as well as aging.[48 49]

The mean age of onset in our cases was 46 years for ALS, 53 years for ALS-parkinsonism, and 46 years for parkinsonism. Thus, these subjects were older than the previously reported cases of ALS (33 years) and parkinsonism (43 years).[1] In our cases, the mean duration of subjective illness was 7.0 years for ALS, 6.3 years for ALS-parkinsonism, and 6.3 years for parkinsonism. The duration of ALS in our data was longer than the world mean of 3 years[21–26] and the previously reported 3.5 years for ALS in Papua.[1] The mean age of onset may have recently increased, and the mean disease duration lengthened in subjects with ALS. The prevalence rate of cases with bulbar sign was 37% in 30 cases of pure ALS or ALS-parkinsonian in our report, similar to the rate reported by Gadjdusek (40%) and of classical ALS (39%).[2] The change in onset and course of ALS in recent years might be due to the aging population and changing environmental and socioeconomic factors.

On-going follow-up surveys should be performed to evaluate changes in the clinical types of ALS and/or parkinsonism with or without CI and accompanying changes in environmental and socioeconomic factors, as have occurred in Guam and Kii.

In conclusion, the prevalence of ALS in high-incidence areas in Papuan has decreased over the past ~30–35 years but, in 2010, was still higher than the global average. During this period, there has been an influx of migrants from inside Papua and transmigrants from outside Papua that has increased the size and changed the composition of the population in the villages studied. This expanded population may have had an impact on prevalence estimates. There are still many cases of pure ALS and overlapping ALS-parkinsonism and CI in Papua, as has previously been reported in Guam and Kii. The disease continues to affect native Papuans, but may also affect a small number of non-Papuan immigrants (two of 46 or 3–4%) who adopt the local lifestyle. Environmental factors may play important aetiological roles. Further on-going follow-up surveys focusing on the clinical epidemiology and aetiology of these neurodegenerative diseases in association with ecological and environmental factors in Papua are needed. Future studies, including neuropathological examinations, should be performed when the local villagers in this region of Papua, Indonesia, are willing to consent to autopsies.

**Author affiliations**
[1]Research Institute for Humanity and Nature, Kyoto, Japan
[2]Center for Southeast Asian Studies, Kyoto University, Kyoto, Japan
[3]Yasugi Clinic, Shimane, Japan
[4]Center for Regional & International Collaboration, Kochi University, Kochi, Japan
[5]Kitaakita Municipal Hospital, Akita, Japan

[6]Faculty of Health Science, Suzuka University of Medical Science, Mie, Japan
[7]Graduate School of Regional Innovation Studies, Mie University, Mie, Japan
[8]Faculty of Medicine, Kochi University, Kochi, Japan
[9]Hakubi Center for Advanced Research, Kyoto University, Kyoto, Japan
[10]Wamena Public Hospital, Papua, Indonesia
[11]Faculty of Medicine, Cenderawasih University, Papua, Indonesia
[12]Faculty of Public Health, Cenderawasih University, Papua, Indonesia

**Acknowledgements** The authors wish to thank all participants in the villages who volunteered for the survey, the village leaders and Mappi Regency officials. We are deeply indebted to doctors Imelda Mandagi, Reginald Nangoy and Christha Z Tamburian and their staff at Puskesmas Bade, and to Arifin Wasaraka, Lazarus Revasy, Yosefina Griapon, Arius Togodly, Kerry Yarangga and Dolfinus Y Bouway at Cenderawasih University for their invaluable contributions to the study. Andre Liem and Herry Wondiwoy at Papua Tour Guides Community and SE Amos at Bade High School provided logistic and language assistance. On the Japanese side, we acknowledge the technical support and advice of Katsuyuki Eguchi (Nagasaki University), Yasushi Osaki (Kochi University), Hidekazu Tomimoto (Mie University), Taro Yamauchi (The University of Tokyo), Hideyuki Onishi (Doshisha Women's College), Tetsuya Inamura (Aichi Prefectural University), Wenling Chen, Yumi Kimura, Yasuko Ishimoto and Hissei Imai (Kyoto University), and Yasuyuki Kosaka, Yukiko Kita and Chizu Wada (Research Institute for Humanity and Nature).

**Contributors** KO had full access to all the data in the study and takes responsibility for the integrity of the data and the accuracy of the data analysis. KO and KM: study concept and design, and study supervision; KO, TW, MF, MI, EGdSH, SK, YK, HS, RS, IM and KM: acquisition of data; KO, TW, RS, MF and KM: analysis and interpretation of data and drafting of the manuscript; KO: statistical analysis; KO, TW, SK, YK and KM: obtaining funding; KO, PW, ALR and KM: administrative, technical or material support; all authors: critical revision of the manuscript for important intellectual content.

**Funding** KO receives research support from Grant-in-Aid for Scientific Research (A) from the Japan Society for the Promotion of Science (JSPS) (21256005, 25257507). KM receives research support from Grant-in-Aid for Scientific Research (A) from the JSPS (23241079) and is an editor of the journal Geriatrics & Gerontology International. TW receives research support from Grant-in-Aid for Scientific Research (C) from the JSPS (21590689). YK and SK receive support from Grants-in-Aid from the Research Committee of CNS Degenerative Diseases and the Research Committee on Muro disease (Kii amyotrophic lateral sclerosis/Parkinsonism-dementia), the Ministry of Health, Labour and Welfare of Japan (200936192, 201024015, 201128085, 201231159).

**Competing interests** None.

**Patient consent** Obtained.

**Ethics approval** The Research Institute for Humanity and Nature in Japan and Cenderawasih University in Papua approved this study.

**Provenance and peer review** Not commissioned; externally peer reviewed.

**Data sharing statement** No additional data are available.

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
