## [Reviewer comments · BMJ Open]

Some articles will have been accepted based in part or entirely on reviews undertaken for other BMJ Group journals. These will be reproduced where possible.

ARTICLE DETAILS

TITLE (PROVISIONAL)	Amyotrophic lateral sclerosis and parkinsonism in Papua, Indonesia: 2001–2012 survey results
AUTHORS	Okumiya, Kiyohito; Wada, Taizo; Fujisawa, Michiko; Ishine, Masayuki; Garcia del Saz, Eva; Hirata, Yutaka; Kuzuhara, Shigeki; Kokubo, Yasumasa; Seguchi, Harumichi; Sakamoto, Ryota; Manuaba, Indrajaya; Watofa, Paulina; Rantetampang, Andreas; Matsubayashi, Kozo

VERSION 1 - REVIEW

REVIEWER	Peter Spencer, PhD, FANA, FRCPath Oregon Health & Science University, USA Continued research interest in western Pacific ALS/PDC.
REVIEW RETURNED	18-Nov-2013

GENERAL COMMENTS	GENERAL This is broad epidemiological survey conducted across time in a region of Papua, Indonesia in which indigenous residents have historically experienced a very high incidence of ALS and parkinsonism. While the area covered by the survey is geographically extensive, it does not encompass the totality of villages studied by Gajdusek and Salazar (1982) in 1974, 1976, 1978, 1980 but does include all villages studied in 1987 and 1990 by Spencer et al., 2005. It is unclear from the present survey whether the historical prevalence data includes Gajdusek and Salazar's "confirmed" ALS cases or all of their "confirmed", "probable" and "possible" cases of ALS. While the two above-mentioned studies focused on essentially sessile populations, in more recent years there may have increased movement of people from affected villages and immigration of non-native people from other parts of Indonesia. These points should be mentioned. INTRODUCTION Correction: The former name of Papua, Indonesia is Irian Jaya, Indonesia. Explain that the Iri river is in the epicenter(s) of the high-incidence focus of ALS and parkinsonism. METHODS State how the Japanese neurologists and Indonesian investigators communicated with the subjects studied. Most subjects probably would not have spoken English or Indonesian. Who served as translators and how was their reliability in translation established? How was written informed consent obtained in this highly illiterate population? Given a belief among some Auyu people that touching the subject may exacerbate an illness such as ALS, how was permission obtained to conduct a neurological examination?
--

How was cognitive function assessed? State the diagnostic criteria for “poliomyeloradiculitis” as originally described by Gajdusek.

Case Reports

Use of the word “ethnicity” is questionable. The people in Papua are described as Papuans, Melanesians and Austronesians; the authors should state if the first group is the dominant or even exclusive group in the population studied. The Auyu refers to a linguistic grouping; these people are probably best described in this way, i.e. the Auyu linguistic group.

Use of the word “allegedly” is also questioned. Who made the allegation, the subject, a family member, other? A more precise description might be written thus: “the subject with ALS reported that his father showed (specific features), which were interpreted to be consistent with parkinsonism.”

Several of the case reports refer to subjects residing in villages at or near the top of rivers; these villages (e.g. Bosuma, Pies) are less visited, least developed, and tended historically to have the largest number of cases. This was evident from an analysis of data generated by Gajdusek [1] who provided village-by-village data showing the number of cases of ALS, parkinsonism and “poliomyeloradiculitis”. A comparable analysis would permit a determination of whether there was a trend for the more remote villages to show a greater change in prevalence of ALS or parkinsonism. The authors should examine their village-specific data to compare and contrast with Gajdusek’s earlier findings and to determine whether there was a relationship between distance up the river and disease prevalence.

DISCUSSION

The discussion should begin with a description of the population, the environment, and what changes have occurred over time. Some of this appears in a later paragraph. The reader should be informed at outset that the population was composed of former hunter-gatherers who lived in the forest, and who were brought out of the forest and assembled in villages by Dutch colonialists. Mention should be made of the existence of neurodegenerative disease before the introduction of an manufactured item, as emphasized by Gajdusek [1]. The reader should also be informed that Indonesia acquired the western portion of New Guinea and, with the former nation’s transmigration policy, Indonesians have in recent years settled in Papua; this has resulted in various changes, including the use of food and medicine. Additionally, the authors should comment on the state of nutrition of subjects and their ALS/PDC-unaffected family members.

p.12. Were the 97 cases reported by Gajdusek and Salzar considered to be “confirmed” cases?

p.13. Given that ALS prevalence was found to be 3 times lower than found by Gajdusek, the statement “ALS may have decreased during the 35 years” is overly conservative. The statement should be changed to indicate that the findings are consistent with those of Spencer et al. (2005), who showed a decline in the prevalence of ALS in the population of the river Ia. This is the key finding of the present study and, secondarily, that disease prevalence is still higher than the global average.

p.15. The sentence “However, the responsible gene(s) remain unproved in almost all cases of ALS/PDC in both Guam and Kii.” stands in stark contrast to the statement in the Introduction that the high incidence of ALS/PDC in Guam and Kii reportedly no longer exists, which clearly indicates that the disease is dominated by, or is exclusively, non-genetic in origin. This conclusion is strengthened by the three obviously diverse genotypes (Chamorro, Japanese, Papuan) affected by ALS/PDC. Based on their new data from Papua, and their uncontested recognition of published data from Kii and Guam (all showing the declining prevalence), the authors should make a clear statement affirming the conclusions of Gajdusek and Spencer that disease etiology in Papua must be dominated by environmental factors. Additionally, based on (1) Gajdusek’s observation that ALS and parkinsonism had not spread beyond the focus area (which might suggest an infectious agent), coupled with (2) his failure experimentally to transmit ALS/PDC from humans to primates (unlike his prior successful results with kuru) [1], Gajdusek concluded that a slow virus (later named “prion”) was not responsible for ALS/PDC. The sentence referring to prions and epigenetic factors (for which there is no evidence) should be deleted.

p.14. Etiology is also briefly mentioned in paragraph #2 on p.14. While the references are adequate, the discussion is superficial. Gajdusek emphasized the etiology was almost certainly environmental but could not involve any manufactured product because the disease was present in Papua before their introduction. He supported the Japanese proposal (originally based on studies in Kii) that the riverine communities surveyed here suffered from mineral imbalance because of low concentrations of calcium and magnesium in drinking water. Spencer et al. (2005) pointed out that this explanation was inconsistent with a declining prevalence of ALS, which the present work has documented.

Spencer et al. (1987) reported the discovery that cycad seed was used as a poultice in Papua, as in Guam, and later showed that exposure to raw cycad seed was a common factor in Papua (poultice) and Kii (oral tonic) cases of ALS, in addition to epidemiological evidence from Guam showing that traditional use of cycad seed in young adulthood was significantly associated with later-life neurodegenerative disease. While these “links are so far unproven”, their observation should not be dismissed, as in paragraph #2, p14, without description of available evidence.

p.16. The second paragraph is useful but omits the key question, namely whether the traditional medicinal practice of using cycad poultices continues, since Spencer et al. (2005) identified this as a putative etiologic factor. Sago eaten by the Auyu and Jaqai, it should be mentioned, is prepared from the trunk of Metroxylon sp. (true sago palm) not the overground stem of cycad species (false sago palm).

The study also omits important clinical observations, namely whether subjects with ALS or parkinsonism showed (1) evidence of prior skin wounds (traditionally treated with a cycad poultice), (2) skin changes as reported by Ono and colleagues, and (3) retinal pigment epitheliopathy, a very high incidence of which has been documented in Guam ALS/PDC and is reportedly present in some Japanese cases. These are serious omissions since findings may be helpful in illuminating the relationship between the disease in Papua

	and that in Guam and Kii Peninsula, and in the potentially common etiology. Comment is also needed on: (1) the gender distribution in relation to that in Guam and Kii cases; (2) the mean age of parkinsonism relative to the mean age of ALS, which was found to be lower in Papua than in Guam and Kii; (3) whether any cases of pure or dominant “dementia” were identified, equivalent to the cases of so-called “Guam dementia” (thought to be the last stage of this disappearing disease), and (4) since the authors appear to have encountered cases of “poliomyelorradiculitis”, their description and interpretation of this ill-defined illness would be useful. Finally, the term “case suffered” (text and table legends) should be changed to “active cases” or something equivalent, and the term “parkinsonian hands” needs definition. ABSTRACT. Objective. Only one prior follow-up study of ALS and parkinsonism has been carried out (Spencer et al.). This was not an incidence study. Redraft this paragraph without stating the name(s) of previous author(s). Conclusions. Change the emphasis from high disease prevalence to confirmation that the prevalence has declined. Strengths and Limitations. Change the second paragraph from an emphasis on high prevalence to confirmation that the prevalence has declined. Address whether this population has been sessile over the past 50 years, and whether there has been an influx of Indonesian transmigrants over the past 10-15 years that has changed the population total in the villages studied. Reference 1. Gajdusek, D.C. Foci of motor neuron disease in high incidence in isolated populations of east Asia and the western Pacific. In Rowland LP, ed, Human Motor Neuron Diseases, pp363-393. See Figure 4. The major conclusion to be drawn from the present study is that the prevalence of ALS has declined, which confirms the study of Spencer et al. (2005), which focused on a component of the population studied here. If disease prevalence is declining in accord with the experience of other similar high-incidence foci (Guam, Kii Peninsula, Honshu Island), this supports the contention of both Gajdusek and Spencer that environmental factors must have an important etiologic role. Unfortunately, these points are not discussed and, instead, the authors’ main conclusion is simply that high incidence disease continues to be present in the population studied.
--	--

REVIEWER	Walter G. Bradley University of Miami Miller School of Medicine, Miami, FL. USA
REVIEW RETURNED	29-Nov-2013

GENERAL COMMENTS	This is a virtually unique paper about the Papua focus of ALS/PDC. No significant followup has been reported since the original report of this by Gadjusek in the 1980's. There have been important followups of the foci in Guam and the Kii Peninsula, but none of the Papua focus. This paper fills this gap excellently.
--

	The authors are to be congratulated on an excellent and important study. They are correct in stating that neuropathological studies will be important if and when the villagers agree to autopsies. The authors are encouraged: 1. to undertake further genetic studies to determine if there is familial linkage of the cases and if any causative or predisposing genes can be identified; and 2. to undertake detailed environmental and ethnological studies to search for potential causative environmental factors.
--	--

REVIEWER	Glen Kisby Associate Professor of Pharmacology Department of Basic Medical Sciences Director, COMP-Northwest Research Lab Western University of Health Sciences
REVIEW RETURNED	15-Dec-2013

GENERAL COMMENTS	General: The major goal of this paper is to determine the prevalence of ALS and parkinsonism in the south coastal region of Papua, Indonesia. Cases of ALS and parkinsonism were determined by well trained neurologists based solely on established clinical signs of these diseases with or without cognitive impairment (CI). While the focus of the paper is clearly on ALS and Parkinsonism in this region, the significance of CI in either group of cases (~20% of the 46 cases) was poorly described. These findings are novel, but there are several issues with the interpretation of the data that dampens the significance of their findings. The following concerns were noted in the manuscript: Major Concerns: 1. Abstract (p. 2): The authors state that their follow-up study of the incidence of ALS and Parkinsonism in Papua, Indonesia (present paper) is the only one since the survey studies by Gajdusek and colleagues. Yet, Spencer et al. (2005) conducted follow-up studies on the same region of Indonesia described in the paper. The authors need to revise their introductory statement by deleting “since a survey undertaken from 1962 to 1980 by Gajdusek and colleagues.” 2. Abstract (p.2): The authors conclude that the prevalence of ALS is lower than previously reported (vs. Gajdusek and/or Spencer). If their conclusion is based on comparison to one or both studies, then a more correct interpretation would be that the prevalence of ALS has declined. They also conclude that ALS/Parkinsonism is higher than that previously reported for Guam and Kii. Missing from their conclusions is a description of the significance of the CI in either ALS or ALS/P cases. 3. Introduction (p. 4): The authors state “there have been few follow-up surveys on ALS and Parkinsonism in Papua since 1991.” This would imply that additional surveys were done. If this is the case, they also need to be described and cited, otherwise change ‘few’ to ‘no’. 4. Results (p. 7): The authors give an overview of the ALS, Parkinsonism and ALS/P cases and those with overlapping CI for the 46 cases in Papua. Upon follow-up (next paragraph), changes in both groups (ALS and ALS/P) were observed, but there is no mention of the CI cases. Did any of these cases also have
--

	overlapping CI? Were there any increases in CI in either the ALS or ALS/P groups? If they only observed changes in a few ALS and ALS/P cases, then the authors should also state that the CI cases did not change. 5. Discussion (p. 12, lines 39-45): The authors state that they found comparable cases with clinical features of ALS, Parkinsonism, ALS-P with and without dementia as did the previous survey by Gadjusek and colleagues. Are their findings also comparable to the survey conducted by Spencer and colleagues (2005)? Since the authors did not report neurological assessment of the overlapping cases of CI (n= 9 cases) on follow-up, they cannot conclude that they have dementia. A more accurate description would be that they are cognitively impaired as described later on in the discussion (p. 14, lines 19-32). 6. Discussion (p.13, lines 27-30): The authors state that the prevalence of ALS may have decreased during the 35 years since Gadjusek's and colleagues report. There is no mention of the survey by Spencer or colleagues (2005). They need to include this information in their conclusions. As stated before (see Abstract changes above), a more accurate description would be to state that there has been a decline in prevalence. 7. Discussion (p. 14, lines 43-52). The significance of their findings in Papua to the prevalence of ALS and Parkinsonism on Guam/Kii is poorly addressed. The authors underscored the significance of environmental factors, but later on indicate that they might have an important role (p. 15, lines 48-52). They even conclude that future follow-up studies should address accompanying changes in environmental factors (lines 36-43). More importantly, the authors omitted mentioning two papers by Borenstein and colleagues (2007) that documents strong associations of cycad exposure with mild cognitive impairment, dementia and PDC among Guam cases. They should at least acknowledge that such associations have been found, especially since CI was documented in both ALS and ALS-P cases in their survey. Minor Changes: 1. Results: (p. 9, line 5): Since this case had a diagnosis of CI, they need to add + CI to heading to distinguish between this case and the previous one (i.e., ALS 12). 2. Results: (p. 10, line 41): Since this case had a diagnosis of CI, then need to add + CI to heading to distinguish between this ALS-P case and the previous one (i.e., ALS-P2).
--	--

REVIEWER	Douglas Galasko UC San Diego USA
REVIEW RETURNED	24-Dec-2013

GENERAL COMMENTS	An interesting study of ALS and Parkinsonism in an area of New Guinea where high-incidence ALS and Parkinson-Dementia complex were described decades ago. The authors are to be commended on their diligence in surveying a number of villages in this area and performing detailed Neurological evaluations of patients to ascertain clinical syndromes/diagnoses. They appear to
---

	have identified a large number of patients with ALS and Parkinson's disease. A few questions need to be clarified by the authors:  1. How were cases identified within each village? Were only suspected cases examined, or were people above a certain age also briefly examined as well? 2. What is the age structure (population pyramid) of the villages that were surveyed? If life expectancy is relatively low and few people live beyond the age of 50, for example, then the overall prevalence figure will be an underestimate if appropriate age-adjustment is carried out. 3. Similarly, what is the sex (gender) distribution across the villages surveyed? It would be interesting to know whether male gender is a risk factor for ALS and Parkinsonism or not, and some idea of this can be obtained from the denominator. 4. Calculating prevalence per 100,000 from a base population that is much lower than 100,000 is complicated. The authors should discuss whether a confidence interval can be assigned to their overall prevalence estimate. Also, because their surveys were carried out over > 1 year, they should distinguish between the approximate prevalence rates they are presenting and true point prevalence. 5. The discussion should comment on the clinical course of ALS in these villages in New Guinea. It seems to be considerably slower than that of typical ALS. 6. The authors should acknowledge the limitations of not having EMG data, DNA analysis, or any autopsy data from this population. Without such data, it is difficult to be certain how these patients fit into sporadic ALS/PD and how they compare with ALS/PD on Guam and in the Kii peninsula.
--	---

VERSION 1 – AUTHOR RESPONSE

Dear Dr Peter Spencer

Thank you very much for your precise comments and I revised as follows.

(Reply-1 to the general comment)

We discussed the historical prevalence distinguishing clinically confirmed cases from all cases including those not clinically confirmed, because there is the possibility that possible cases in El Escorial criteria may be included in the cases not clinically confirmed. Crude point prevalence rate of ALS consistent with clinically probable to definite ALS (El Escorial criteria) was estimated between clinically confirmed cases and all cases.

The underlined revised sentences are described in the discussion.

Gajdusek et al. reported 13 cases of ALS and three cases of parkinsonism in 1975 in the villages along the Ia River (population, 2000).[1] The 13 ALS cases consisted of 10 clinically confirmed ones and 3 others that were not clinically confirmed.

A crude point prevalence of ALS consistent with clinically probable to definite ALS (El Escorial criteria) may be estimated between 500 (95% confidence interval, 191–809/100,000 people) and 650 (298–1002) and that of parkinsonism was 150 (0–320/100,000 people) in 1975 along the Ia River.

Gajdusek et al. reported 8 cases of ALS and 6 cases of parkinsonism in 1975 in the villages along the Edera and Dumut rivers (population, 1950). The 8 cases of ALS consisted of 7 clinically confirmed cases and 1 unclinically confirmed case. The crude point prevalence of ALS consistent with clinically probable to definite ALS (El Escorial criteria) may be estimated between 359 (94–624)/100,000 people and 410 (127–694)/100,000 people, and that of parkinsonism was 308 (62–554)/100,000 people in 1975 along the Edera and Dumut rivers.

(Reply-2 to the general comment) Before the discussion of the change of prevalence, the change of the population in the field sites should be discussed. The population of Irian Jaya increases from 923,000 in 1971 to 1730,000 in 1990 and the percentage of people born outside Irian Jaya increase from 4 to 21 % (urban:4 to 12%, rural 0.3 to 9%).[16] In the recent 10 years the population of former Irianjaya (Province of Papua and Papua Barat) has increased by 1.6 times (2.2 million in 2000, 3.6 million in 2010) by the massive migration and increase of Papuan people. [18]

The underlined revised sentences are described in the discussion.

The population of Bade increased by a factor of 4 during 31 years (from 1000 in 1975 to 3800 in 2006), and those of villages along the Ia River (from 2000 to 4500) and the Edera and Dumut rivers (from 1950 to 4100) have increased 2-fold.[1,15] The subjects born outside Papua were only 2 (Maluku) among the 46 subjects in this survey. The prevalence of the diseases among the local Papuan residents in this survey may be underestimated compared with the former report after taking into account the increase in the numbers of local migrants inside and transmigrants from outside Papua.[16,17,18]

INTRODUCTION

Correction: The former name of Papua, Indonesia is Irian Jaya, Indonesia. Explain that the Ia river is in the epicenter(s) of the high-incidence focus of ALS and parkinsonism.

The underlined revised sentences are described in the introduction.

Between 1962 and 1980, Gajdusek et al. reported 97 cases of amyotrophic lateral sclerosis (ALS), 18 cases of parkinsonism, and 18 cases of poliomyeloradiculitis (PMR) in the south coastal area of Papua (former Irian Jaya, Indonesia), which had a population of 7,000. The Ia River is in the epicenter of the high-incidence focus of ALS and parkinsonism in Papua.[1,2]

METHODS

State how the Japanese neurologists and Indonesian investigators communicated with the subjects studied. Most subjects probably would not have spoken English or Indonesian. Who served as translators and how was their reliability in translation established? How was written informed consent obtained in this highly illiterate population? Given a belief among some Auyu people that touching the subject may exacerbate an illness such as ALS, how was permission obtained to conduct a neurological examination?

The underlined revised sentences are described in the METHODS.

Indonesian collaborators and the co-authors (an English teacher in Bade Senior High School and Indonesian neurologist IM, and staff from Cenderawasih University) who spoke both English and Indonesian; local people who spoke both Indonesian and the local language in each village; and co-author EG, who spoke English, Indonesian, and Japanese, cooperatively served as translators. Written informed consent was obtained from all the study participants with the support of their family members. All the participants agreed to undergo neurological examinations, and none of the patients declined annual re-examination.

How was cognitive function assessed?

The underlined revised sentences are described in the METHODS.

CI was diagnosed when loss of memory or impairments of language usage, praxis, or executive functions were identified during interview and clinical examination by neurologists. As cognitive functional test such as the mini-mental state examination was not performed for all of the participants, patients with complications of functional disabilities attributable to CI in activities of daily living or in their livelihood were screened during the interview of both the patients and their family members.

The underlined revised sentences are described in the DISCUSSION.

CI might have been underestimated, and mild CIs or some mild dementia might have been overlooked by our neurological examinations in the field.

State the diagnostic criteria for “poliomyelorradiculitis” as originally described by Gajdusek.

The underlined revised sentences are described in the method.

PMR was evaluated using the definition of Gajdusek et al. that PMR is a subacute paralytic condition reminiscent of the Landry-Guillain-Barre syndrome, which combines radicular, or perhaps neuritic elements with more acute onset and sometimes asymmetric paralysis.[1,2] Neurological cases with overlapping cerebellar signs were excluded from this study.

Case Reports

Use of the word “ethnicity” is questionable. The people in Papua are described as Papuans, Melanesians and Austronesians; the authors should state if the first group is the dominant or even exclusive group in the population studied. The Auyu refers to a linguistic grouping; these people are probably best described in this way, i.e. the Auyu linguistic group.

(Reply) The word “ethnicity” was revised to be tribe.

Use of the word “allegedly” is also questioned. Who made the allegation, the subject, a family member, other? A more precise description might be written thus: “the subject with ALS reported that his father showed (specific features), which were interpreted to be consistent with parkinsonism.”

The following document are added in the result.

The subject had ALS, and his brother reported that their father showed tremor and gait disturbance, which were interpreted to be consistent with parkinsonism.

Her younger sister was case P 6. The subject had ALS-parkinsonism, and her sister reported that their cousin showed tremor and gait disturbance, which were interpreted to be consistent with parkinsonism (Figure 3B).

Several of the case reports refer to subjects residing in villages at or near the top of rivers; these villages (e.g. Bosuma, Pies) are less visited, least developed, and tended historically to have the largest number of cases. This was evident from an analysis of data generated by Gajdusek [1] who provided village-by-village data showing the number of cases of ALS, parkinsonism and “poliomyelorradiculitis”. A comparable analysis would permit a determination of whether there was a trend for the more remote villages to show a greater change in prevalence of ALS or parkinsonism. The authors should examine their village-specific data to compare and contrast with Gajdusek’s earlier findings and to determine whether there was a relationship between distance up the river and

disease prevalence.

The underlined revised sentences are described in the RESULT.

In the survey during the period 2007–2012, the number of each clinical type (ALS, ALS-parkinsonism, and parkinsonism) was shown respectively for Bade (4, 4, and 1); the villages along the Ia River, namely Ogorito (1, 2, and 0), Bosma (1, 1, and 0), Homlikya (1, 1, and 1), Gimikya (1, 1, and 0), Geturki (1, 0, and 0), Asset (1, 0, and 1), and Osso (1, 0, and 1); and the villages along the Edera and Dumut rivers, namely Yibin (1, 1, and 0), Sien (0, 1, and 2), Pies (1, 0, and 1), Benggo (0, 1, and 0), Harapan (0, 0, and 1), Mopio (0, 0, and 1), Memes (0, 0, and 1), Muya (1, 0, and 0), and Getio (0, 1, and 0).

The underlined revised sentences are described in the DISCUSSION.

In the report by Gajdusek et al., who provided village-by-village data showing the number of cases of ALS and parkinsonism, there were many cases in the remote villages such as Bosma, Pies, Homlikya, Yibin, Sien, and Ogorite, which were less visited and least developed.[1, 2] In this survey, many cases (three ALS/ALS-parkinsonism) were found in the remotest village of Ogorito along the Ia River. Two cases of ALS/ALS-parkinsonism were also found in Bosma, Homlikya, and Yibin; 1 ALS and 2 cases of parkinsonism in Sien; 1 ALS and 1 parkinsonism in Pies, which consisted also of remote villages. While there were 8 cases of ALS/ALS-parkinsonism in Bade and 2 in Gimikya, nearest to the port of the Digul River, none was reported by Gajdusek et al. in those sites. This increase may be brought about by local migrants from the affected villages.

DISCUSSION

The discussion should begin with a description of the population, the environment, and what changes have occurred over time. Some of this appears in a later paragraph. The reader should be informed at outset that the population was composed of former hunter-gatherers who lived in the forest, and who were brought out of the forest and assembled in villages by Dutch colonialists. Mention should be made of the existence of neurodegenerative disease before the introduction of an manufactured item, as emphasized by Gadjusek [1]. The reader should also be informed that Indonesia acquired the western portion of New Guinea and, with the former nation's transmigration policy, Indonesians have in recent years settled in Papua; this has resulted in various changes, including the use of food and medicine. Additionally, the authors should comment on the state of nutrition of subjects and their ALS/PDC-unaffected family members.

The underlined revised sentences are described in the DISCUSSION.

Internal migration and migration from outside the province are the most significant causes of demographic, social, and cultural change in Papua. The populations in the survey sites were composed of former hunter-gatherers who lived in the forest and who were brought out of the forest and assembled in villages by Dutch colonialists. The villages along the Ia and Digul rivers were founded in 1940–1950.[10] The Dutch colonial program of population distribution from highly populated regions such as Java and Bali to Papua and other less populated province was continued by the Indonesian Government. The emigration program called transmigration (transmigrasi) started in Irian Jaya in 1964. Local migrants and transmigrants from outside Irian Jaya increased, resulting in various changes, including the use of food and medicine.[16,17] Considering that Gadjusek et al. found out the high prevalence of ALS at the first survey in 1962, the neurodegenerative disease existed before the introduction of manufactured items.[1]

p.12. Were the 97 cases reported by Gajdusek and Salzar considered to be “confirmed” cases?

The underlined revised sentences are described in DISCUSSION.

Gajdusek et al. reported diagnosing 97 cases (57 cases clinically confirmed and 40 cases not clinically confirmed) of ALS during the 6 surveys they conducted from 1962 to 1980.[1]

p.13. Given that ALS prevalence was found to be 3 times lower than found by Gajdusek, the statement “ALS may have decreased during the 35 years” is overly conservative. The statement should be changed to indicate that the findings are consistent with those of Spencer et al. (2005), who showed a decline in the prevalence of ALS in the population of the river Ia. This is the key finding of the present study and, secondarily, that disease prevalence is still higher than the global average.

The following sentences are revised in the DISCUSSION.

The prevalence of ALS in this report was found to be three times lower than found by Gajdusek et al. Even after taking into account the change of population and age structure in the field sites, our findings show that the prevalence of ALS is supposed to be decreased during the 35 years since the report of Gajdusek et al. This is consistent with the previous report by Spencer et al. [10] However, it is still much higher than the global mean (incidence, approximately 2/100,000; prevalence, approximately 6/100,000) [21-26]; in particular, the prevalence of overlapping ALS and parkinsonism is especially high in Papua.

p.15. The sentence “However, the responsible gene(s) remain unproved in almost all cases of ALS/PDC in both Guam and Kii.” stands in stark contrast to the statement in the Introduction that the high incidence of ALS/PDC in Guam and Kii reportedly no longer exists, which clearly indicates that the disease is dominated by, or is exclusively, non-genetic in origin. This conclusion is strengthened by the three obviously diverse genotypes (Chamorro, Japanese, Papuan) affected by ALS/PDC. Based on their new data from Papua, and their uncontested recognition of published data from Kii and Guam (all showing the declining prevalence), the authors should make a clear statement affirming the conclusions of Gajdusek and Spencer that disease etiology in Papua must be dominated by environmental factors. Additionally, based on (1) Gajdusek’s observation that ALS and parkinsonism had not spread beyond the focus area (which might suggest an infectious agent), coupled with (2) his failure experimentally to transmit ALS/PDC from humans to primates (unlike his prior successful results with kuru) [1], Gajdusek concluded that a slow virus (later named “prion”) was not responsible for ALS/PDC. The sentence referring to prions and epigenetic factors (for which there is no evidence) should be deleted.

The underlined revised sentences are described in the DISCUSSION.

As the diseases are declining as in Guam and Kii, the disease aetiology in Papua may be dominated by environmental factors. Genetic factors may also play a role as evidenced by the family history of disease in 40% of the subjects in this survey in Papua and 80% in Kii.

(Reply) The sentence referring to prions and epigenetic factors were deleted.

p.14. Etiology is also briefly mentioned in paragraph #2 on p.14. While the references are adequate, the discussion is superficial. Gadjusek emphasized the etiology was almost certainly environmental but could not involve any manufactured product because the disease was present in Papua before their introduction. He supported the Japanese proposal (originally based on studies in Kii) that the riverine communities surveyed here suffered from mineral imbalance because of low concentrations of calcium and magnesium in drinking water. Spencer et al. (2005) pointed out that this explanation was inconsistent with a declining prevalence of ALS, which the present work has documented.

The underlined revised sentences are described in the DISCUSSION.

There is a recent report showing an association between the increase in ALS incidence and the change in drinking water source in the Kii Peninsula.[35] Spencer et al. pointed out that this explanation was inconsistent with a declining prevalence of ALS in spite of drinking water from shallow wells, which this report may support.[10]

Spencer et al. (1987) reported the discovery that cycad seed was used as a poultice in Papua, as in Guam, and later showed that exposure to raw cycad seed was a common factor in Papua (poultice) and Kii (oral tonic) cases of ALS, in addition to epidemiological evidence from Guam showing that traditional use of cycad seed in young adulthood was significantly associated with later-life neurodegenerative disease. While these “links are so far unproven”, their observation should not be dismissed, as in paragraph #2, p14, without description of available evidence.

The underlined revised sentences are described in the DISCUSSION.

Spencer et al. reported the discovery that cycad seed was used as a poultice in Papua, as in Guam. [36] Cycad-derived products were shown as the risk factor of dementia, mild CI, and PDC in Guam with the epidemiological evidence.[37] Although cycad had been suspected as the causal factor of Kii ALS,[38] people in Japan, including the Kii Peninsula, had never consumed cycad as daily food, although a small amount of cycad flour might have been taken by a few people as an ingredient of herb medicine.[39]

p.16. The second paragraph is useful but omits the key question, namely whether the traditional medicinal practice of using cycad poultices continues, since Spencer et al. (2005) identified this as a putative etiologic factor. Sago eaten by the Auyu and Jaqai, it should be mentioned, is prepared from the trunk of Metroxylon sp. (true sago palm) not the overground stem of cycad species (false sago palm).

The underlined revised sentences are described in the DISCUSSION.

Indigenous people of Papua still retain aspects of their traditional lifestyles, such as hunting, gathering, fishing, and eating sago (trunk of Metroxylon sp.).

The study also omits important clinical observations, namely whether subjects with ALS or parkinsonism showed (1) evidence of prior skin wounds (traditionally treated with a cycad poultice), (2) skin changes as reported by Ono and colleagues, and (3) retinal pigment epitheliopathy, a very high incidence of which has been documented in Guam ALS/PDC and is reportedly present in some Japanese cases. These are serious omissions since findings may be helpful in illuminating the relationship between the disease in Papua and that in Guam and Kii Peninsula, and in the potentially common etiology.

The underlined revised sentences are described in the DISCUSSION.

We did not examine the association of prior skin wounds and exposure to cycad poultice with the diseases. Skin change and pigmentary retinopathy were not examined in this survey.[40, 41] Further studies are needed.

Comment is also needed on: (1) the gender distribution in relation to that in Guam and Kii cases; (2) the mean age of parkinsonism relative to the mean age of ALS, which was found to be lower in Papua than in Guam and Kii; (3) whether any cases of pure or dominant “dementia” were identified, equivalent to the cases of so-called “Guam dementia” (thought to be the last stage of this

disappearing disease), and (4) since the authors appear to have encountered cases of "poliomyelorradiculitis", their description and interpretation of this ill-defined illness would be useful. Finally, the term "case suffered" (text and table legends) should be changed to "active cases" or something equivalent, and the term "parkinsonian hands" needs definition.

(1) the gender distribution in relation to that in Guam and Kii cases;

The underlined revised sentences are described in the discussion.

The proportion of males in the populations of Bade, the villages along the Ia River, and the villages along the Edera and Dumut rivers were 53%, 51%, and 50%. The proportion of males was high especially among the subjects with ALS-parkinsonism (85%) and ALS (59%) compared with among those with pure parkinsonism (50%) in our report. The proportion of males was higher among the subjects with ALS (63%) and parkinsonism (76%) in the report by Gadjusek et al., in which there were 3 male cases (75%) among the 4 ALS-parkinsonism cases. The high risks for ALS and PDC among the males were recognized in Guam and Kii.[1,7,27]

(2) the mean age of parkinsonism relative to the mean age of ALS, which was found to be lower in Papua than in Guam and Kii;

(Reply)

The mean age of onset and the duration of ALS became older in this report compared with the report by Gadjusek, so the mean age of parkinsonism relative to the mean age of ALS, which was found to be lower in Papua.

The underlined revised sentences are described in the discussion.

The mean age of onset in our cases was 46 years for ALS, 53 years for ALS-parkinsonism, and 46 years for parkinsonism. Thus, they were older than the previously reported cases of ALS (33 years) and parkinsonism (43 years).[1] In our cases, the mean duration of subjective illness was 7.0 years for ALS, 6.3 years for ALS-parkinsonism, and 6.3 years for parkinsonism. The duration of ALS in our data was longer than not only the world mean of 3 years[21-26] but also the previously reported 3.5 years for ALS in Papua.[1] The mean age of onset may recently have increased, and the mean disease duration became longer in subjects with ALS. The prevalence rate of cases with bulbar sign was 37% in 30 of pure ALS or ALS-parkinsonian in our report, and it was as the same as in the report of Gadjusek (40%) and classical ALS (39%).[2] The change in onset and course of ALS in recent years might be attributable to the aging population and changing environmental and socioeconomic factors.

(3) whether any cases of pure or dominant "dementia" were identified, equivalent to the cases of so-called "Guam dementia" (thought to be the last stage of this disappearing disease)

(4) since the authors appear to have encountered cases of "poliomyelorradiculitis", their description and interpretation of this ill-defined illness would be useful.

The underlined revised sentences are described in the result.

We found no case of pure or dominant dementia and PMR consistent with the definition by Gadjusek et al.

"case suffered" (text and table legends) should be changed to "active cases" or something equivalent, and the term "parkinsonian hands" needs definition.

(Reply) "Case suffered" (text and table legends) was changed to "active cases".

The following document are added in the figure legend.

Parkinsonian hand: Striatal deformities of the hand with abnormal postures that are common in patients with advanced Parkinson's disease.

ABSTRACT. Objective. Only one prior follow-up study of ALS and parkinsonism has been carried out (Spencer et al.). This was not an incidence study. Redraft this paragraph without stating the name(s) of previous author(s). Conclusions. Change the emphasis from high disease prevalence to confirmation that the prevalence has declined.

The underlined revised sentences are described in the ABSTRACT.

Only one prior follow-up study of amyotrophic lateral sclerosis (ALS) and parkinsonism in Papua, Indonesia has been carried out since a survey undertaken from 1962 to 1980 by Gajdusek and colleagues.

The prevalence of ALS should have decreased during the 35 years since Gajdusek et al.'s report. However, it remains higher than the global average.

Strengths and Limitations. Change the second paragraph from an emphasis on high prevalence to confirmation that the prevalence has declined. Address whether this population has been sessile over the past 50 years, and whether there has been an influx of Indonesian transmigrants over the past 10-15 years that has changed the population total in the villages studied.

The underlined revised sentences are described in the strength.

- The prevalence of ALS is supposed to be decreased since Gajdusek et al.'s report during the 35 years, when there has been an influx of migrants and transmigrants from outside Papua that has changed the population total in the villages studied. However, the prevalence is still higher than the global average.

Reference

1. Gajdusek, D.C. Foci of motor neuron disease in high incidence in isolated populations of east Asia and the western Pacific. In Rowland LP, ed, Human Motor Neuron Diseases, pp363-393. See Figure 4.

(Reply) The reference was revised.

The major conclusion to be drawn from the present study is that the prevalence of ALS has declined, which confirms the study of Spencer et al. (2005), which focused on a component of the population studied here. If disease prevalence is declining in accord with the experience of other similar high-incidence foci (Guam, Kii Peninsula, Honshu Island), this supports the contention of both Gajdusek and Spencer that environmental factors must have an important etiologic role. Unfortunately, these points are not discussed and, instead, the authors' main conclusion is simply that high incidence disease continues to be present in the population studied.

The following document are revised in the conclusion.

In conclusion, The prevalence of ALS is supposed to be decreased during the 35 years since the report of Gajdusek et al., when there has been an influx of migrants inside and transmigrants from outside Papua that has changed the population in the villages studied. However, there are still many cases of pure ALS and overlapping ALS-parkinsonism and CI, as has previously been reported in

Guam and Kii. Environmental factors may play important etiologic role.

Dear Dr Walter G. Bradley

Thank you very much for your precise comments and I revised as followings.

Revised to be "None declared"

Dear Dr Glen Kisby

Thank you very much for your precise comments and I revised as followings.

Revised to be "None declared"

Major Concerns:

1. Abstract (p. 2): The authors state that their follow-up study of the incidence of ALS and Parkinsonism in Papua, Indonesia (present paper) is the only one since the survey studies by Gajdusek and colleagues. Yet, Spencer et al. (2005) conducted follow-up studies on the same region of Indonesia described in the paper. The authors need to revise their introductory statement by deleting "since a survey undertaken from 1962 to 1980 by Gajdusek and colleagues."

The underlined revised sentences are described in the ABSTRACT.

Only one prior follow-up study of amyotrophic lateral sclerosis (ALS) and parkinsonism in Papua, Indonesia has been carried out since a survey undertaken from 1962 to 1980 by Gajdusek and colleagues.

2. Abstract (p.2): The authors conclude that the prevalence of ALS is lower than previously reported (vs. Gajdusek and/or Spencer). If their conclusion is based on comparison to one or both studies, then a more correct interpretation would be that the prevalence of ALS has declined. They also conclude that ALS/Parkinsonism is higher than that previously reported for Guam and Kii. Missing from their conclusions is a description of the significance of the CI in either ALS or ALS/P cases.

The underlined revised sentences are described in the conclusion.

The prevalence of ALS should have decreased during the 35 years since Gajdusek et al.'s report. However, it remains higher than the global average. There was a high prevalence of overlap of ALS, parkinsonism, and CI, which has been previously reported in Guam and Kii.

3. Introduction (p. 4): The authors state "there have been few follow-up surveys on ALS and Parkinsonism in Papua since 1991." This would imply that additional surveys were done. If this is the case, they also need to be described and cited, otherwise change 'few' to 'no'.

The underlined revised sentences are described in the INTRODUCTION.

However, no follow-up surveys on ALS and parkinsonism have been conducted in Papua since 1991.

4. Results (p. 7): The authors give an overview of the ALS, Parkinsonism and ALS/P cases and those with overlapping CI for the 46 cases in Papua. Upon follow-up (next paragraph), changes in both groups (ALS and ALS/P) were observed, but there is no mention of the CI cases. Did any of these cases also have overlapping CI? Were there any increases in CI in either the ALS or ALS/P groups? If they only observed changes in a few ALS and ALS/P cases, then the authors should also state that the CI cases did not change.

The following sentences are revised in the RESULT.

During the follow-up surveys of the cases in the Bade, Ia, and Edera rivers, we observed the following changes in diagnosis: 1 case, from possible ALS to probable ALS (case ALS10); 2 cases, from parkinsonism to ALS-parkinsonism (cases ALS-P 2 and ALS-P 3); and 1 case, from probable ALS to definite ALS (case ALS16) complicated with CI.

5. Discussion (p. 12, lines 39-45): The authors state that they found comparable cases with clinical features of ALS, Parkinsonism, ALS-P with and without dementia as did the previous survey by Gadjusek and colleagues. Are their findings also comparable to the survey conducted by Spencer and colleagues (2005)? Since the authors did not report neurological assessment of the overlapping cases of CI (n= 9 cases) on follow-up, they cannot conclude that they have dementia. A more accurate description would be that they are cognitively impaired as described later on in the discussion (p. 14, lines 19-32).

The following sentences are revised in the METHOD.

CI was diagnosed when loss of memory or impairments of language usage, praxis, or executive functions were identified during interview and clinical examination by neurologists. As cognitive functional test such as the mini-mental state examination was not performed for all of the participants, patients with complications of functional disabilities attributable to CI in activities of daily living or in their livelihood were screened during the interview of both the patients and their family members.

The following sentences are revised in the DISCUSSION.

Spencer et al reported 2 cases of ALS and three cases of parkinsonism, including 1 with dementia, in the survey in 1987. He also reported 3 subjects with parkinsonism overlapping UMN signs, including 1 with dementia in 1991. We have here reported 17 cases of pure ALS (probable or definite), 13 cases of overlapping ALS-parkinsonism, and 16 cases of parkinsonism with or without CI in the 46 cases we neurologically examined fully. Thus, we identified many cases of overlapping ALS and parkinsonism (n = 13, 28%) in our study and in previous reports (n = 6, 23% and n = 3, 38%).[1, 10] We also identified many cases with CI (n = 9, 20%) in our survey and in previous reports (n = 7, 27% and n = 2, 25%).[1, 10] Functional disabilities attributable to CI were recognized in activities of daily living or in their livelihood in all the cases with CI in this report.

The underlined revised sentences are described in the CONCLUSION.

The prevalence of CI was lower in our study sites than in Guam and Kii.[4, 27-33] CI might have been underestimated, and mild CIs or some mild dementia might have been overlooked by our neurological examinations in the field.

6. Discussion (p.13, lines 27-30): The authors state that the prevalence of ALS may have decreased during the 35 years since Gadjusek's and colleagues report. There is no mention of the survey by Spencer or colleagues (2005). They need to include this information in their conclusions. As stated before (see Abstract changes above), a more accurate description would be to state that there has been a decline in prevalence.

The following sentences are revised in the DISCUSSION.

The prevalence of ALS in this report was found to be three times lower than found by Gajdusek et al. Even after taking into account the change of population and age structure in the field sites, our findings show that the prevalence of ALS is supposed to be decreased during the 35 years since the report of Gajdusek et al. This is consistent with the previous report by Spencer et al.

7. Discussion (p. 14, lines 43-52). The significance of their findings in Papua to the prevalence of ALS and Parkinsonism on Guam/Kii is poorly addressed. The authors underscored the significance of environmental factors, but later on indicate that they might have an important role (p. 15, lines 48-52). They even conclude that future follow-up studies should address accompanying changes in environmental factors (lines 36-43). More importantly, the authors omitted mentioning two papers by Borenstein and colleagues (2007) that documents strong associations of cycad exposure with mild cognitive impairment, dementia and PDC among Guam cases. They should at least acknowledge that such associations have been found, especially since CI was documented in both ALS and ALS-P cases in their survey.

The following sentences are revised in the DISCUSSION.

Spencer et al. reported the discovery that cycad seed was used as a poultice in Papua, as in Guam. [36] Cycad-derived products were shown as the risk factor of dementia, mild CI, and PDC in Guam with the epidemiological evidence.[37]

Minor Changes:

1. Results: (p. 9, line 5): Since this case had a diagnosis of CI, they need to add + CI to heading to distinguish between this case and the previous one (i.e., ALS 12).
2. Results: (p. 10, line 41): Since this case had a diagnosis of CI, then need to add + CI to heading to distinguish between this ALS-P case and the previous one (i.e., ALS-P2).

(Reply) CI was added in the two cases.

Dear Dr Douglas Galasko

Thank you very much for your precise comments and I revised as followings.

Revised to be "None declared"

In some areas, editing by a native English speaker would be helpful.

(Reply) This documents was revised by native speaker too.

An interesting study of ALS and Parkinsonism in an area of New Guinea where high-incidence ALS and Parkinson-Dementia complex were described decades ago. The authors are to be commended on their diligence in surveying a number of villages in this area and performing detailed Neurological evaluations of patients to ascertain clinical syndromes/diagnoses. They appear to have identified a large number of patients with ALS and Parkinson's disease. A few questions need to be clarified by the authors:

1. How were cases identified within each village? Were only suspected cases examined, or were

people above a certain age also briefly examined as well?

The underlined revised sentences are described in the method.

The purpose of the neurological examinations was explained to all of the villagers. All the subjects volunteered to participate after the announcement of availability of neurological check-ups, which were performed in the community health centres or homes in each field site. While this was not a complete inventory survey, we asked each medical doctor or health staff, and village leader to call all patients with neurological signs or symptoms, including muscle weakness, gait disturbance, tremor, bradykinesia, or cognitive impairment (CI), in each village. The patients who were called also volunteered to participate. Indonesian collaborators and the co-authors (an English teacher in Bade Senior High School and Indonesian neurologist IM, and staff from Cenderawasih University) who spoke both English and Indonesian; local people who spoke both Indonesian and the local language in each village; and co-author EG, who spoke English, Indonesian, and Japanese, cooperatively served as translators. Written informed consent was obtained from all the study participants with the support of their family members. All the participants agreed to undergo neurological examinations, and none of the patients declined annual re-examination.

2. What is the age structure (population pyramid) of the villages that were surveyed? If life expectancy is relatively low and few people live beyond the age of 50, for example, then the overall prevalence figure will be an underestimate if appropriate age-adjustment is carried out.

(Reply)

The median age of Papua province in 2010 was 22.8 years, which was younger than average of Indonesia (27.2), global average(28.5), Guam (29.2) and Japan (44.9). [18] The life expectancy (both sex) of Indonesia in 2010 was 67.9 years, which was the same as global average (67.9) and lower than Guam (75.5) and Japan (82.7). [19] As the life expectancy of field sites in Papua may be lower than global average and fewer people may live beyond the age of 40 or 50, the prevalence of the diseases in this survey may be underestimated compared with global average. Median age of people in 1997 in Edera district including Bade, Ia and Edera& Dumut was between 15-19, which was very young. [20] As the median age and life expectancy in 1975 may be much lower than those in 2010, the prevalence of the diseases in the report by Gajdusek et al might be underestimated compared with that of our survey.

The underlined revised sentences are described in the method.

As the median age and life expectancy in the field sites in Papua are estimated to be lower than the global median and fewer people may live beyond the age of 40 or 50 years,[18,19] the prevalence of the diseases in this survey may be underestimated compared with the global mean. As the median age and life expectancy in 1975 may be lower than those in 2010,[20] the prevalence of the diseases in the report by Gajdusek et al. might be underestimated compared with that in our survey.

3. Similarly, what is the sex (gender) distribution across the villages surveyed? It would be interesting to know whether male gender is a risk factor for ALS and Parkinsonism or not, and some idea of this can be obtained from the denominator.

The underlined revised sentences are described in the discussion.

The proportion of males in the populations of Bade, the villages along the Ia River, and the villages along the Edera and Dumut rivers were 53%, 51%, and 50%. The proportion of males was high especially among the subjects with ALS-parkinsonism (85%) and ALS (59%) compared with among those with pure parkinsonism (50%) in our report. The proportion of males was higher among the subjects with ALS (63%) and parkinsonism (76%) in the report by Gadusek et al., in which there were 3 male cases (75%) among the 4 ALS-parkinsonism cases. The high risks for ALS and PDC among

the males were recognized in Guam and Kii.[1,7,27]

4. Calculating prevalence per 100,000 from a base population that is much lower than 100,000 is complicated. The authors should discuss whether a confidence interval can be assigned to their overall prevalence estimate. Also, because their surveys were carried out over > 1 year, they should distinguish between the approximate prevalence rates they are presenting and true point prevalence.

(Reply) 95% confidence interval were calculated and described in text and table 4. Crude point prevalence rate was revised to be approximate crude point prevalence rate.

5. The discussion should comment on the clinical course of ALS in these villages in new Guinea. It seems to be considerably slower than that of typical ALS.

The underlined revised sentences are described in the discussion. In our cases, the mean duration of subjective illness was 7.0 years for ALS, 6.3 years for ALS-parkinsonism, and 6.3 years for parkinsonism. The duration of ALS in our data was longer than not only the world mean of 3 years[21-26] but also the previously reported 3.5 years for ALS in Papua.[1] The mean age of onset may recently have increased, and the mean disease duration became longer in subjects with ALS. The prevalence rate of cases with bulbar sign was 37% in 30 of pure ALS or ALS-parkinsonian in our report, and it was as the same as in the report of Gajdusek (40%) and classical ALS (39%).[2] The change in onset and course of ALS in recent years might be attributable to the aging population and changing environmental and socioeconomic factors.

6. The authors should acknowledge the limitations of not having EMG data, DNA analysis, or any autopsy data from this population. Without such data, it is difficult to be certain how these patients fit into sporadic ALS/PD and how they compare with ALS/PD on Guam and in the Kii peninsula.

The underlined revised sentences are described in Strengths and limitations of this study and the discussion.

This study was based only on clinical findings and there are the limitations of not having electromyogram data, DNA analysis, or any autopsy data from this population. Without such data, it is difficult to be certain how these patients fit into sporadic ALS or Parkinson's disease and how they compare with ALS/PDC on Guam and in the Kii peninsula.

VERSION 2 – REVIEW

REVIEWER	Peter Spencer, PhD, FANA, FRCPath Oregon Health & Science University, USA Author conducts research on the etiology of western Pacific ALS-PDC, the subject of this paper.
REVIEW RETURNED	22-Feb-2014

GENERAL COMMENTS	GENERAL The revised paper with track changes marked was used to complete this review. There is repeated reference to the 35-year period between Gajdusek's observations and those of the present study. Gajdusek made preliminary observations of high-incidence motor neuron disease (ALS) among Jaqai and Auyu-speaking people in 1962 (Gajdusek, New Engl J Med 268:474-6, 1963). He conducted village surveys beginning in April-May 1974 (see Table 1 in Gajdusek's 1979 chapter in Amyotrophic Lateral Sclerosis, Tsubaki
---

T, Toyokura Y, eds, University of Tokyo Press, Tokyo, 1979:287-303). The time period between the Gajdusek's data-collection period and that of the present (2010) study is thus 36 years. However, Gajdusek carried out surveys through 1981, which amounts to only 29 years between the two data-collection periods. It is therefore recommended that reference is made to a "~30-35-year period" instead of a "35-year period" wherever this descriptor is used in the manuscript.

ABSTRACT

Conclusions

CHANGE TO: "While the prevalence of ALS has decreased over the past ~30-35 years, it remains higher than the global average."

Strengths and Limitations

DELETE the second bullet – it is neither a strength nor a limitation of this study.

INTRODUCTION

Line 12. DELETE the word "However,"

RESULTS

Page 8, Line 1. CHANGE TO: "We found no case of pure or dominant dementia and no case of PMR consistent with the definition of Gajdusek et al.; similarly, no cases of PMR were found by Spencer et al. in 1987 or 1990."

DISCUSSION

THE ACCURACY OF THE FOLLOWING STATEMENT IS QUESTIONED. "The Dutch colonial program of population distribution from highly populated regions such as Java and Bali to Papua and other less populated province was continued by the Indonesian Government."

The authors should clarify this statement to ensure historical and geographical accuracy: The Dutch liberated Indonesia from their colonial control in 1949. Papua was almost completely populated by indigenous Papuans in 1960. The Indonesian (not Dutch) transmigration program to Papua (Irian Jaya) commenced circa 1985, although 1500 transmigration plots had been abandoned by 1986. Transmigration during the 1980s was faster than in the 1990s. Eventually, transmigrants to Papua comprised approximately half of the total population of the province. To what extent these statements apply to the populations studied in the present project is unknown, but it is likely they were proportionately much less heavily impacted by transmigration than less remote areas of Papua.

http://www.academia.edu/1196557/Transmigration_in_Indonesia_Lessons_from_its_environmental_and_social_impacts

Figure 1 legend of Spencer al. (2005) states the dates of village creation along the river Ila from 1937 (Gimikya) to 1951 (Bosuma), not "1940-1950", as stated by the authors..

Page 14, paragraph 2. REPLACE "1962-1980" WITH "1974-1981." See explanation above.

Page 15, first paragraph. CHANGE "1991" TO "1990".

Page 15. Last paragraph, line 5. CHANGE "present" TO "recent"

Page 16. If correct, make it clear that of the 46 subjects with ALS+parkinsonism+CI, 44 were of Papuan genetic stock and 2 (from Maluku) were non-Papuan.

Page 17, line 11. REPLACE "is supposed to be" WITH "appears to have"

Page 20, line 2. ALS-PDC has appeared in high incidence in three genetically disparate populations (Japanese, Chamorro and Papuan New Guinean). It is difficult to argue that genetics is a major etiologic factor for ALS in Guam because disease incidence has

	dramatically declined to near-worldwide levels. Additionally, ALS-PDC on Guam has been recognized in a few genetically distinct non-Chamorro immigrants who adopted the Chamorro lifestyle, comparable perhaps to the 2 of 46 cases that likely adopted the local lifestyle of the Auyu/Jaqai people after their migration the Moluccas (Maluku) to Papua. Since the presence of disease in a family across time can suggest hereditary factors, culture-associated behaviors, family-specific practices, or mixtures thereof, it is strongly recommended that the authors REPLACE “Genetic” WITH “Genetic, culture-related and/or family-specific factors”. Page 20, paragraph 2. REPLACE “in spite of drinking water from shallow wells” WITH “in a sessile population dependent on an unchanging supply of river water” Page 22. REPLACE “The prevalence of ALS is supposed to be decreased during the 35 years since the report of Gajdusek et al., when there has been an influx of migrants inside and transmigrants from outside Papua that has changed the population in the villages studied.” WITH “The prevalence of ALS in the Papuan disease focus has decreased over the past ~30-35 years but, in 2010, it remained higher than the global average. During this period, there has been an influx of migrants from inside and transmigrants from outside Papua that has increased the size and changed the composition of the population in the villages studied.” Should the conclusion comment on? (a) the possible impact of the expanded population on prevalence estimates, and (b) the disease continues to impact native Papuans but may also affect a small number of non-Papuan immigrants (2 of 46 or 3-4%) who adopt the local lifestyle. The authors have inserted data (as previously requested) on the changing population (size and composition), which may affect prevalence calculations. The authors have responded to most of the concerns but minor problems and some imprecision remain. Several of the concerns detailed below understandably relate to authors with a non-native command of written English. It is recommended that the second and hopefully final revision of this manuscript responds to the points below and is then edited by a native English writer.
--	--

REVIEWER	Glen Kisby Associate Professor of Pharmacology Western University of Health Sciences United State of America (USA)
REVIEW RETURNED	27-Feb-2014

GENERAL COMMENTS	General: The authors have responded adequately to questions regarding CI in the previous version of the manuscript. However, there are still minor changes that need to be completed before the manuscript can be accepted for publication. Minor changes (from version with track changes):  1. Abstract (Conclusions): The first two sentences are contradictory. Delete the words 'should have' in the first sentence and combine with second sentence. 2. Abstract (Strengths & Limitations): Delete the second bullet. Suggest replacing with the finding that significant overlap of ALS with other neurodegenerative diseases was noted. 3. Methods (p.44, 1sr paragraph): The number of individuals who
--

	were excluded due to overlapping cerebellar signs should be stated. Presumably, this was a very small number relative to the 46 cases that they evaluated for ALS, Parkinsonism and CI. 4. Discussion: (p. 52, lines 27-28). Delete 'in our study and' and Change to 'like that observed'. 5. Discussion: (p. 52, lines 29-30). Delete 'in our survey and' and Change to 'like that observed'. 6. Discussion: (p. 52, lines 49-50). Change 'and' to 'with'. 7. Discussion: (p. 53, lines 20-21). Change 'unclinically confirmed' to 'unconfirmed'. 8. Discussion: (p. 54, lines 30-31). Delete 'is supposed to be'. Either the prevalence increased of decreased. It can't be both. 9. Discussion: (p. 59, lines 45-46). Delete 'is supposed to be'. Either the prevalence increased of decreased. It can't be both.
--	---

REVIEWER	Douglas Galasko UCSD, USA
REVIEW RETURNED	27-Feb-2014

GENERAL COMMENTS	A few minor edits are needed to improve the English. For example, prevalence 'is supposed to be decreased' should be reworded, e.g., 'was expected to have decreased' would be better in the Abstract, and 'was reported to have decreased' would be more appropriate in the discussion of the repeated survey by Spencer. This could be done by the authors with assistance from BMJ Editorial staff The authors have responded to the questions raised by reviewers.
---

VERSION 2 – AUTHOR RESPONSE

GENERAL

The revised paper with track changes marked was used to complete this review.

There is repeated reference to the 35-year period between Gajdusek's observations and those of the present study. Gajdusek made preliminary observations of high-incidence motor neuron disease (ALS) among Jaqai and Auyu-speaking people in 1962 (Gajdusek, *New Engl J Med* 268:474-6, 1963). He conducted village surveys beginning in April-May 1974 (see Table 1 in Gajdusek's 1979 chapter in *Amyotrophic Lateral Sclerosis*, Tsubaki T, Toyokura Y, eds, University of Tokyo Press, Tokyo, 1979:287-303). The time period between the Gajdusek's data-collection period and that of the present (2010) study is thus 36 years. However, Gajdusek carried out surveys through 1981, which amounts to only 29 years between the two data-collection periods. It is therefore recommended that reference is made to a "~30-35-year period" instead of a "35-year period" wherever this descriptor is used in the manuscript.

The sentence in objective in the abstract was revised as following.

Only one prior follow-up study of amyotrophic lateral sclerosis (ALS) and parkinsonism in Papua, Indonesia has been carried out since a survey undertaken from 1962 to 1981 by Gajdusek and colleagues.

ABSTRACT

Conclusions

CHANGE TO: "While the prevalence of ALS has decreased over the past ~30-35 years, it remains higher than the global average."

Conclusion in the abstract was revised as following.

While the prevalence of ALS has decreased over the past ~30-35 years, it remains higher than the global average.

Strengths and Limitations

DELETE the second bullet – it is neither a strength nor a limitation of this study.

The second bullet was deleted.

INTRODUCTION

Line 12. DELETE the word "However,"

"However," was deleted.

RESULTS

Page 8, Line 1. CHANGE TO: "We found no case of pure or dominant dementia and no case of PMR consistent with the definition of Gajdusek et al.; similarly, no cases of PMR were found by Spencer et al. in 1987 or 1990."

The sentence in the result was revised as following.

We found no case of pure or dominant dementia and PMR consistent with the definition by Gajdusek et al; similarly, no cases of PMR were found by Spencer et al. in 1987 or 1990. [10]

DISCUSSION

THE ACCURACY OF THE FOLLOWING STATEMENT IS QUESTIONED. "The Dutch colonial program of population distribution from highly populated regions such as Java and Bali to Papua and other less populated province was continued by the Indonesian Government."

The authors should clarify this statement to ensure historical and geographical accuracy:

The Dutch liberated Indonesia from their colonial control in 1949. Papua was almost completely populated by indigenous Papuans in 1960. The Indonesian (not Dutch) transmigration program to Papua (Irian Jaya) commenced circa 1985, although 1500 transmigration plots had been abandoned by 1986. Transmigration during the 1980s was faster than in the 1990s. Eventually, transmigrants to Papua comprised approximately half of the total population of the province. To what extent these statements apply to the populations studied in the present project is unknown, but it is likely they were proportionately much less heavily impacted by transmigration than less remote areas of Papua.

http://www.academia.edu/1196557/Transmigration_in_Indonesia_Lessons_from_its_environmental_and_social_impacts

The sentence in the DISCUSSION was deleted.

Figure 1 legend of Spencer al. (2005) states the dates of village creation along the river Ia from 1937 (Gimikya) to 1951 (Bosuma), not "1940-1950", as stated by the authors..

The sentence was revised as following.

The villages along the Ia River were founded in 1937–1951.[10]

Page 14, paragraph 2. REPLACE “1962-1980” WITH “1974-1981.” See explanation above.

The sentence was revised as following.

Gajdusek et al. reported diagnosing 97 cases (57 cases clinically confirmed and 40 cases not clinically confirmed) of ALS during the 6 surveys they conducted from 1974 to 1981.[1]

Page 15, first paragraph. CHANGE “1991” TO “1990”.

The sentence was revised as following.

He also reported 3 subjects with parkinsonism overlapping UMN signs, including 1 with dementia in 1990.

Page 15. Last paragraph, line 5. CHANGE “present” TO “recent”

The sentence was revised as following.

We compared the prevalence of probable-definite ALS and parkinsonism according to the report of Gajdusek et al. with our more recent findings.

Page 16. If correct, make it clear that of the 46 subjects with ALS+parkinsonism+CI, 44 were of Papuan genetic stock and 2 (from Maluku) were non-Papuan.

The sentence was revised as following.

The subjects born outside Papua were only 2 (Maluku) among the 46 subjects with ALS and/or parkinsonism with/without CI in this survey, ie 44 were of Papuan genetic stock and 2 (from Maluku) were non-Papuan.

Page 17, line 11. REPLACE “is supposed to be” WITH “appears to have”

The sentence was revised as following.

Even after taking into account the change of population and age structure in the field sites, our findings show that the prevalence of ALS appears to have decreased over the past ~30-35 years since the report of Gajdusek et al.[10]

Page 20, line 2. ALS-PDC has appeared in high incidence in three genetically disparate populations (Japanese, Chamorro and Papuan New Guinean). It is difficult to argue that genetics is a major etiologic factor for ALS in Guam because disease incidence has dramatically declined to near-worldwide levels. Additionally, ALS-PDC on Guam has been recognized in a few genetically distinct non-Chamorro immigrants who adopted the Chamorro lifestyle, comparable perhaps to the 2 of 46 cases that likely adopted the local lifestyle of the Auyu/Jaqai people after their migration the Moluccas (Maluku) to Papua.

Since the presence of disease in a family across time can suggest hereditary factors, culture-associated behaviors, family-specific practices, or mixtures thereof, it is strongly recommended that

the authors REPLACE “Genetic” WITH “Genetic, culture-related and/or family-specific factors”.

The sentence was revised as following.

Genetic, culture-related and/or family-specific factors may also play a role as evidenced by the family history of disease in 40% of the subjects in this survey in Papua and 80% in Kii.

Page 20, paragraph 2. REPLACE “in spite of drinking water from shallow wells” WITH “in a sessile population dependent on an unchanging supply of river water”

The sentence was revised as following.

Spencer et al. pointed out that this explanation was inconsistent with a declining prevalence of ALS in a sessile population dependent on an unchanging supply of river water, which this report may support.[10]

Page 22. REPLACE “The prevalence of ALS is supposed to be decreased during the 35 years since the report of Gajdusek et al., when there has been an influx of migrants inside and transmigrants from outside Papua that has changed the population in the villages studied.” WITH “The prevalence of ALS in the Papuan disease focus has decreased over the past ~30-35 years but, in 2010, it remained higher than the global average. During this period, there has been an influx of migrants from inside and transmigrants from outside Papua that has increased the size and changed the composition of the population in the villages studied.”

The sentence was revised as following.

The prevalence of ALS in the Papuan disease focus has decreased over the past ~30-35 years but, in 2010, it remained higher than the global average. During this period, there has been an influx of migrants from inside and transmigrants from outside Papua that has increased the size and changed the composition of the population in the villages studied.

Should the conclusion comment on? (a) the possible impact of the expanded population on prevalence estimates, and (b) the disease continues to impact native Papuans but may also affect a small number of non-Papuan immigrants (2 of 46 or 3-4%) who adopt the local lifestyle.

The authors have inserted data (as previously requested) on the changing population (size and composition), which may affect prevalence calculations.

These sentences were described in the conclusion.

There may be the possible impact of the expanded population on prevalence estimates.

The disease continues to impact native Papuans but may also affect a small number of non-Papuan immigrants (2 of 46 or 3-4%) who adopt the local lifestyle.

The authors have responded to most of the concerns but minor problems and some imprecision remain. Several of the concerns detailed below understandably relate to authors with a non-native command of written English. It is recommended that the second and hopefully final revision of this manuscript responds to the points below and is then edited by a native English writer.

The manuscript has already been revised by a native English writer.

Dear Dr Glen Kisby

Thank you very much for your precise comments and I revised as followings.

Please state any competing interests or state 'None declared':

Revised to be "None declared"

General:

The authors have responded adequately to questions regarding CI in the previous version of the manuscript. However, there are still minor changes that need to be completed before the manuscript can be accepted for publication.

Minor changes (from version with track changes):

1. Abstract (Conclusions): The first two sentences are contradictory. Delete the words 'should have' in the first sentence and combine with second sentence.

Conclusion in the abstract was revised as following.

While the prevalence of ALS has decreased over the past ~30-35 years, it remains higher than the global average.

2. Abstract (Strengths & Limitations): Delete the second bullet. Suggest replacing with the finding that significant overlap of ALS with other neurodegenerative diseases was noted.

The sentence was described in Strengths & Limitations.

This study recognized the clinical findings of significant overlap of ALS with parkinsonism and cognitive impairment.

3. Methods (p.44, 1st paragraph): The number of individuals who were excluded due to overlapping cerebellar signs should be stated. Presumably, this was a very small number relative to the 46 cases that they evaluated for ALS, Parkinsonism and CI.

The sentence was revised as following.

Neurological cases with overlapping cerebellar signs (2 cases) were excluded from this study.

4. Discussion: (p. 52, lines 27-28). Delete 'in our study and' and Change to 'like that observed'.

The sentence was revised as following.

Thus, we identified many cases of overlapping ALS and parkinsonism (n = 13, 28%) like that observed in previous reports (n = 6, 23% and n = 3, 38%).[1, 10]

5. Discussion: (p. 52, lines 29-30). Delete 'in our survey and' and Change to 'like that observed'.

The sentence was revised as following.

We also identified many cases with CI (n = 9, 20%) like that observed in previous reports (n = 7, 27% and n = 2, 25%).[1, 10]

6. Discussion: (p. 52(15), lines 49-50). Change 'and' to 'with'.

We compared the prevalence of probable-definite ALS and parkinsonism according to the report of

Gajdusek et al. with our more recent findings.

7. Discussion: (p. 53(16), lines 20-21). Change 'unclinically confirmed' to 'unconfirmed'.

The sentence was revised as following.

The 8 cases of ALS consisted of 7 clinically confirmed cases and 1 unconfirmed case.

8. Discussion: (p. 54(17), lines 30-31). Delete 'is supposed to be'. Either the prevalence increased or decreased. It can't be both.

The sentence was revised as following.

Even after taking into account the change of population and age structure in the field sites, our findings show that the prevalence of ALS appears to have decreased over the past ~30-35 years since the report of Gajdusek et al.[10]

9. Discussion: (p. 59(22), lines 45-46). Delete 'is supposed to be'. Either the prevalence increased or decreased. It can't be both.

The sentence was revised as following.

The prevalence of ALS in the Papuan disease focus has decreased over the past ~30-35 years but, in 2010, it remained higher than the global average.

Dear Dr Douglas Galasko

Thank you very much for your precise comments and I revised as followings.

Please state any competing interests or state 'None declared':

Revised to be "None declared"

A few minor edits are needed to improve the English. For example, prevalence 'is supposed to be decreased' should be reworded, e.g., 'was expected to have decreased' would be better in the Abstract, and 'was reported to have decreased' would be more appropriate in the discussion of the repeated survey by Spencer.

This could be done by the authors with assistance from BMJ Editorial staff

The authors have responded to the questions raised by reviewers.

Following the comments of the two reviewers the sentences were revised as below.

While the prevalence of ALS has decreased over the past ~30-35 years, it remains higher than the global average.

Even after taking into account the change of population and age structure in the field sites, our findings show that the prevalence of ALS appears to have decreased over the past ~30-35 years since the report of Gajdusek et al.[10]